# Endogenous control of inflammation characterizes pregnant women with asymptomatic or paucisymptomatic SARS-CoV-2 infection

Sara De Biasi [1,10], Domenico Lo Tartaro [1,10], Lara Gibellini [1,10], Annamaria Paolini[1], Andrew Quong[2], Carlene Petes[2], Geneve Awong[2], Samuel Douglas[2], Dongxia Lin[2], Jordan Nieto[2], Francesco Maria Galassi[3], Rebecca Borella[1], Lucia Fidanza[1], Marco Mattioli[1], Chiara Leone[1], Isabella Neri[1], Marianna Meschiari [4], Luca Cicchetti[5], Anna Iannone[6], Tommaso Trenti[7], Mario Sarti[7], Massimo Girardis[8], Giovanni Guaraldi [4], Cristina Mussini[4], Fabio Facchinetti [1,11] & Andrea Cossarizza [1,9,11✉]

SARS-CoV-2 infection can affect all human beings, including pregnant women. Thus, understanding the immunological changes induced by the virus during pregnancy is nowadays of pivotal importance. Here, using peripheral blood from 14 pregnant women with asymptomatic or mild SARS-CoV-2 infection, we investigate cell proliferation and cytokine production, measure plasma levels of 62 cytokines, and perform a 38-parameter mass cytometry analysis. Our results show an increase in low density neutrophils but no lymphopenia or gross alterations of white blood cells, which display normal levels of differentiation, activation or exhaustion markers and show well preserved functionality. Meanwhile, the plasma levels of anti-inflammatory cytokines such as interleukin (IL)-1RA, IL-10 and IL-19 are increased, those of IL-17, PD-L1 and D-dimer are decreased, but IL-6 and other inflammatory molecules remain unchanged. Our profiling of antiviral immune responses may thus help develop therapeutic strategies to avoid virus-induced damages during pregnancy.

[1] Department of Medical and Surgical Sciences for Children and Adults, University of Modena and Reggio Emilia School of Medicine, Modena, Italy. [2] Fluidigm Corporation, South San Francisco, CA, USA. [3] College of Humanities, Arts and Social Sciences, Flinders University, Bedford Park, SA 5042, Australia. [4] Infectious Diseases Clinics, AOU Policlinico and University of Modena and Reggio Emilia, Modena, Italy. [5] Labospace, Milano, Italy. [6] Department of Surgery, Medicine, Dentistry and Morphological Sciences, University of Modena and Reggio Emilia School of Medicine, Modena, Italy. [7] Department of Clinical Pathology, AOU Policlinico, Modena, Italy. [8] Department of Anesthesia and Intensive Care, AOU Policlinico and University of Modena and Reggio Emilia, Modena, Italy. [9] National Institute for Cardiovascular Research, Bologna, Italy. [10]These authors contributed equally: Sara De Biasi, Domenico Lo Tartaro, Lara Gibellini. [11]These authors jointly supervised: Fabio Facchinetti, Andrea Cossarizza. ✉email: andrea.cossarizza@unimore.it

SARS-CoV-2 infection has rapidly become a public health emergency of international concern culminating in the WHO declaration of the pandemic. Scientists from all over the world are investigating the pathogenetic mechanisms of SARS-CoV-2 action, but still little is known about particular groups of patients. Clinical manifestation of COVID-19 can be characterized by mild or asymptomatic infection of the upper-respiratory tract, infection of the lower-respiratory tract with or without life-threatening pneumonia and eventually acute respiratory distress syndrome[1].

Previous studies have shown a variety of immunological changes in patients of all ages, but limited data are currently available for pregnant women with COVID-19 pneumonia[2], or even with paucisymptomatic or asymptomatic SARS-CoV-2 infection. The rate of mortality in those infected by SARS-CoV-2 was unaffected by pregnant status during the first wave in the United States (pregnant 0.19%, nonpregnant 0.25%)[2], being much lower than that reported by sister coronavirus infections, such as SARS (18%) and Middle East respiratory syndrome (MERS, 25%)[3]. At the same time, the spread of infection in pregnancy was similar to the general population, according to surveillance systems in the United Kingdom[4] and Italy[5]. However, in a recent global evaluation (192 studies) regarding 64,676 pregnant or recently pregnant women with COVID-19, they were more frequently admitted to the intensive care unit and submitted to extracorporeal membrane oxygenation (ECMO) with respect to nonpregnant women of similar age, despite reporting less fever and cough. Poor maternal outcomes were predicted by well-known risk factors, such as older age, obesity, diabetes mellitus, chronic hypertension[6], as well as by laboratory findings such as high levels of serum D-dimer and interleukin-6[4]. The impact on pregnancy seems limited to a small increased risk of indicated preterm birth, since poor maternal/fetal oxygen exchanges, while possible increase of other perinatal complications, such as stillbirth or cesarean-section rate, remains controversial[6,7].

Concerns are now arising with the second/third wave of the pandemic where more severe illness seems to characterize pregnant women with respect to the first wave, independently from COVID-19 strain[8]. Even though COVID-19 manifestation seems to be less severe in pregnant women than in elderly patients, it could be not completely absent, or silent. Therefore, more investigational studies need to be done to assess the true impact of COVID-19 on pregnancy. A recent review of 63 observational studies on a total of 637 women with SARS-CoV-2 infection revealed that more than 3 out of 4 experienced mild disease[7].

In order to better understand the severity of the infection, clinical outcomes on pregnant women with COVID-19 were compared with those of SARS and MERS[3]. Interestingly, case fatality was 0%, 18%, and 25%, respectively. In addition, clinical manifestations reported in pregnant women were mild and similar to those reported in nonpregnant women infected by SARS-CoV-2, with predominant features including fever, cough, dyspnea, and lymphopenia[9]. Vertical transmission has been reported in SARS as ACE2 receptor is widely expressed in the placenta. Even if the structure of the receptor-binding domain of SARS and SARS-CoV-2 is similar, cases of vertical transmission of COVID-19 are possible, although reported in few cases[10–12].

During pregnancy, the immune system undergoes relevant changes, and the immunological shifts that occur in pregnancy are partially related to changes in hormonal levels. The immune system of pregnant woman is characterized by an anti-inflammatory immunological tolerance, and for this reason, many autoimmune diseases go into remission, only to flare again in the early postpartum period[13]. Clearly, this could leave the mother more susceptible to viral infections, as a Th1 response

better helps to contrast viruses[10], even if little is known about the response to SARS-CoV-2[14].

The immune system of pregnant women with SARS-CoV-2 infection is now receiving more attention, and different questions are under deep investigation. Here we show that pregnant women with or without infection are characterized by different plasma levels of pro-inflammatory and anti-inflammatory cytokines. Moreover, PBMC from all women have similar subset distributions and general functions of, and show normal parameters related to cell activation or exhaustion, except for higher amounts of circulating low-density neutrophils in infected pregnant women. A deep characterization of T- and B- lymphocyte subsets, along with monocytes, natural killer, and dendritic cells, shows a substantial lack of effect of the viral infection on such populations.

## Results

### Plasma cytokine levels in COVID-19 pregnant women: dysregulation of a few pro- and anti-inflammatory cytokines and decreased levels of D-dimer. To better characterize the cytokine storm caused by COVID-19 infection, plasma levels of 62 cytokines were measured in 14 infected pregnant women, 28 uninfected and 15 age-matched nonpregnant controls (Table 1).

Regarding growth factors, the levels of platelet derived growth factor (PDGF-AA) and epidermal growth factor (EGF) were higher in pregnant COVID-19 compared with pregnant without COVID-19. PDGF-AA is a plasma factor responsible for vascular remodeling, and recently published data report that COVID-19-infected patients have increased levels of molecules involved in this phenomenon (CD40L, PDGF-AA, and PDGF-AB/BB) that correlate with a high level of the Th2 cytokine IL-4[15]. EGF is involved in cellular proliferation, differentiation, and survival, and can modulate the wound healing response to SARS-CoV[16]. Moreover, COVID-19 infection itself leads to the activation of growth factor receptor signaling, maybe sustaining a more marked EGF production[17]. Besides, the levels of PDGF-AA and EGF correlate with the severity of the disease, and high levels of angiogenesis factors were elevated in hospitalized patients with noncritical COVID-19 infection[18].

The anti-inflammatory status of pregnant women is reflected by the levels of CCL4, CCL5, CCL11, and CXCL1 and CXCL2, which were lower in these women if compared with healthy donors. Such chemokines are able to recruit polymorphonuclear cells, such as monocytes, memory T cells, NK cells, and neutrophils in the site of infection and pregnant COVID-19 women were characterized by higher levels of CCL4, CCL5, and CXCL2 if compared with pregnant women without infection.

Fourteen pro-inflammatory cytokines were then quantified, and similar plasma concentrations were found in pregnant women with or without infection. On the contrary, taking into account anti-inflammatory cytokines, IL-1RA, IL-10, IL-27, and IL-19 were higher in COVID-19 pregnant women if compared with noninfected ones. No differences were found regarding IL-4 and IL-6RA plasma levels.

Higher levels of APRIL and BAFF, fundamental survival factors for different B-cell types, were found in pregnant COVID-19 women compared with those without infection, as well as granzyme B (GRZB), a serine protease released by NK and cytotoxic T cells. The meaning of these data remains to be established, considering that plasmablasts and NK cells had similar proportions in infected or non-infected pregnant women.

PD-L1 belongs to a class of molecules that regulate the balance between protective immunity and host immune-mediated damage: COVID-19 patients showed high plasma levels of

**Table 1 Plasma levels of 62 cytokines measured in 14 infected pregnant women (PP), 28 uninfected pregnant women (PN) and 15 age-matched nonpregnant controls (CTR).**

| | CTR (pg/ml; mean ± SEM) | PN (pg/ml; mean ± SEM) | PP (pg/ml; mean ± SEM) | CTR vs. PN | CTR vs. PP | PN vs. PP |
|---|---|---|---|---|---|---|
| *Growth factors* | | | | | | |
| EGF | 39.3 ± 10.0 | 20.6 ± 4.8 | 50.2 ± 10.0 | ns | ns | $P = 0.003$ |
| FGF basic | 2.08 ± 0.01 | 7.1 ± 1.6 | 6.1 ± 1.6 | $P = 0.01$ | $P = 0.007$ | ns |
| G-CSF | 15.5 ± 2.7 | 17.0 ± 2.9 | 18.9 ± 3.2 | ns | ns | ns |
| GM-CSF | 16.2 ± 2.4 | 18.9 ± 3.0 | 52.6 ± 16.3 | ns | ns | ns |
| PDGF-AA | 2699 ± 720.7 | 1355 ± 262.3 | 2018 ± 418.2 | ns | ns | $P = 0.04$ |
| PDGF-AB/BB | 624.8 ± 152.3 | 527.8 ± 108.2 | 747.7 ± 141.6 | ns | ns | ns |
| TGF-α | 12.8 ± 2.7 | 10.3 ± 2.3 | 14.8 ± 2.7 | ns | ns | ns |
| VEGF | 67.3 ± 8.1 | 72.0 ± 13.1 | 104.0 ± 30.9 | ns | ns | ns |
| IL-3 | 6.3 ± 1.1 | 4.2 ± 0.4 | 6.0 ± 1.2 | ns | ns | ns |
| *Chemokines* | | | | | | |
| CCL2 | 173.2 ± 12.0 | 260.5 ± 94.4 | 220.0 ± 80.4 | ns | ns | ns |
| CCL3 | 20.3 ± 4.2 | 12.7 ± 1.9 | 18.0 ± 2.2 | ns | ns | ns |
| CCL4 | 273.7 ± 42.5 | 145.6 ± 21.6 | 225.5 ± 33.5 | $P = 0.008$ | ns | $P = 0.01$ |
| CCL5 | 37,133 ± 9833 | 12,043 ± 1733 | 30,677 ± 6805 | $P = 0.001$ | ns | $P = 0.01$ |
| CCL11 | 173.7 ± 21.3 | 90.8 ± 11.5 | 113.5 ± 14.9 | $P = 0.001$ | ns | ns |
| CCL19 | 85.8 ± 7.9 | 95.9 ± 13.2 | 123.5 ± 17.8 | ns | ns | ns |
| CCL20 | 20.1 ± 3.8 | 21.1 ± 7.4 | 21.1 ± 4.6 | ns | ns | ns |
| CXCL1 | 92.9 ± 17.6 | 45.1 ± 6.7 | 57.2 ± 10.6 | $P = 0.02$ | ns | ns |
| CXCL2 | 454.2 ± 176.9 | 83.1 ± 14.4 | 296.2 ± 83.0 | $P = 0.0001$ | ns | $P = 0.001$ |
| CXCL10 | 79.6 ± 9.0 | 133.9 ± 22.8 | 814.2 ± 336.7 | ns | ns | ns |
| CX3CL1 | 466.3 ± 49.8 | 423.1 ± 29.9 | 613.4 ± 108.1 | ns | ns | ns |
| *Pro-inflammatory cytokines* | | | | | | |
| IFN-α | 6.6 ± 1.2 | 3.5 ± 2.2 | 4.6 ± 0.6 | $P = 0.03$ | ns | ns |
| IFN-β | 1.9 ± 0.4 | 0.8 ± 0.0 | 0.9 ± 0.0 | $P = 0.001$ | $P = 0.01$ | ns |
| IL-1α | 25.3 ± 5.7 | 9.6 ± 1.4 | 16.3 ± 3.6 | $P = 0.02$ | ns | ns |
| IL-1β | 2.0 ± 0.5 | 2.5 ± 0.9 | 2.7 ± 0.6 | ns | ns | ns |
| IL-6 | 7.8 ± 0.9 | 21.8 ± 10.4 | 22.0 ± 9.1 | ns | ns | ns |
| IL-7 | 2.8 ± 0.6 | 1.7 ± 0.1 | 2.5 ± 0.4 | ns | ns | ns |
| IL-11 | 878.2 ± 64.5 | 722.7 ± 34.5 | 712.2 ± 64.8 | $P = 0.03$ | ns | ns |
| IL-17 | 4.0 ± 0.9 | 2.9 ± 0.3 | 3.0 ± 0.5 | ns | ns | ns |
| IL-17C | 1.7 ± 0.2 | 1.9 ± 0.3 | 2.0 ± 0.3 | ns | ns | ns |
| IL-17E | 15.0 ± 4.8 | 7.8 ± 2.0 | 7.5 ± 1.7 | ns | ns | ns |
| IL-18 | 342.8 ± 36.0 | 586.9 ± 106.2 | 838.3 ± 211.6 | $P = 0.04$ | ns | ns |
| IL-23 | 858.8 ± 71.3 | 967.9 ± 39.8 | 1016 ± 68.8 | ns | ns | ns |
| IL-33 | 11.2 ± 2.4 | 4.6 ± 0.7 | 6.0 ± 1.3 | $P = 0.008$ | ns | ns |
| TNF | 5.3 ± 1.3 | 8.1 ± 1.7 | 8.0 ± 1.1 | ns | ns | ns |
| *Anti-inflammatory cytokines* | | | | | | |
| IL-1RA | 250.7 ± 18.4 | 724.0 ± 159.5 | 2249 ± 652.4 | $P < 0.0001$ | $P < 0.0001$ | $P = 0.004$ |
| IL-4 | 1.5 ± 0.3 | 0.7 ± 0.0 | 1.0 ± 0.1 | $P = 0.008$ | ns | ns |
| IL-6RA | 33,056 ± 1898 | 31,270 ± 851.3 | 33,136 ± 3160 | ns | ns | ns |
| IL-10 | 94.9 ± 26.0 | 56.7 ± 14.5 | 134.8 ± 33.7 | ns | ns | $P = 0.02$ |
| IL-19 | 879.8 ± 149.7 | 433.2 ± 47.3 | 903.4 ± 204.4 | $P = 0.005$ | ns | $P = 0.01$ |
| *Immunoregulatory cytokines* | | | | | | |
| IFN-γ | 20.4 ± 6.7 | 5.0 ± 0.9 | 7.7 ± 1.8 | ns | ns | ns |
| IL-2 | 12.8 ± 3.1 | 4.9 ± 0.8 | 7.7 ± 1.6 | $P = 0.02$ | ns | ns |
| IL-5 | 3.5 ± 0.6 | 1.7 ± 0.1 | 2.0 ± 0.3 | $P = 0.01$ | ns | ns |
| IL-12p70 | 5.1 ± 0.4 | 4.8 ± 0.2 | 5.0 ± 0.0 | ns | ns | ns |
| IL-13 | 56.0 ± 11.0 | 30.7 ± 3.7 | 43.7 ± 6.0 | ns | ns | ns |
| IL-15 | 1.8 ± 0.4 | 1.4 ± 00.2 | 1.6 ± 0.6 | ns | ns | ns |
| IL-27 | 193.3 ± 27.4 | 3521 ± 345 | 1558 ± 465.8 | $P < 0.0001$ | ns | $P = 0.003$ |
| *Other molecules* | | | | | | |
| APRIL | 537.8 ± 78.4 | 427.4 ± 49.2 | 568.5 ± 72.0 | ns | ns | $P = 0.03$ |
| BAFF | 439.9 ± 23.7 | 407.0 ± 21.7 | 502.8 ± 34.2 | ns | ns | $P = 0.01$ |
| BMP2 | 5.6 ± 0.8 | 5.0 ± 0.5 | 5.8 ± 0.6 | ns | ns | ns |
| BMP4 | 12.3 ± 1.5 | 7.9 ± 0.4 | 9.3 ± 0.6 | $P = 0.03$ | ns | ns |
| BMP7 | 17.1 ± 1.7 | 22.1 ± 2.0 | 29.1 ± 4.7 | ns | ns | ns |
| CD40L | 703.5 ± 167.1 | 649.4 ± 104.2 | 774.8 ± 178.2 | ns | ns | ns |
| FAS | 6350 ± 620.0 | 5722 ± 230.0 | 4873 ± 432.9 | ns | ns | $P = 0.02$ |
| FASL | 38.1 ± 2.2 | 38.1 ± 2.0 | 38.8 ± 1.2 | ns | ns | ns |
| FLT-3 | 103.3 ± 8.0 | 111.4 ± 6.0 | 124.3 ± 19.0 | ns | ns | ns |
| GRZB | 8.5 ± 2.0 | 6.4 ± 0.9 | 20.0 ± 6.2 | ns | ns | $P = 0.01$ |
| LEPTIN | 16,936 ± 2558 | 29,470 ± 2801 | 23,304 ± 4603 | $P = 0.006$ | ns | ns |
| LEPTIN R | 31,213 ± 3138 | 49,922 ± 3465 | 42,631 ± 4806 | $P = 0.001$ | ns | ns |
| OPN | 28,742 ± 2376 | 35,965 ± 2547 | 36,601 ± 4836 | ns | ns | ns |
| PD-L1 | 91.0 ± 19.0 | 289.7 ± 19.7 | 197.2 ± 31.9 | $P < 0.0001$ | ns | $P = 0.02$ |
| TACI | 16.9 ± 2.2 | 12.0 ± 1.2 | 13.9 ± 1.1 | ns | ns | ns |
| TRAIL | 48.7 ± 5.8 | 17.6 ± 2.1 | 31.9 ± 7.1 | $P < 0.0001$ | ns | ns |

Data are indicated as mean ± SEM. Two-tailed Mann–Whitney *U*-test with Bonferroni correction has been used. Exact p value is indicated. Source data are provided as a Source Data file.

soluble PD-L1[19]. This molecule plays a remarkable role in pregnancy by inducing maternal immune tolerance to fetal tissue, and is more present in serum of pregnant women compared with nonpregnant ones[20]. However, in our cohort of patients, infected pregnant women showed lower levels of PD-L1. We could

hypothesize that PD-L1 had been used to contrast the action of inflammatory cells, and thus to further protect the fetus.

FAS and FASL belong to the TNF receptor superfamily and play an important role in the regulation of immune homeostasis, and in modulating cell death. The FAS/FASL system contains

both membrane-bound and soluble molecules. It has been reported that soluble FAS blocks apoptosis by inhibiting the binding of FASL to FAS on the cell membrane[21]. We found that FAS was lower in plasma from pregnant COVID-19 women when compared with those without infection. This could likely indicate that, even if the ongoing infection, a decreased plasma level of FAS plays an important role in maternal immunotolerance, as already described in the first trimester of pregnancy[22].

Interestingly, pregnant women positive to SARS-CoV-2 had also significantly lower levels of D-dimer (median value: 785 mg/L) than those who were negative (1640 mg/L; $p = 0.009$; see details in Supplementary data 1). This further reinforces the presence of an anti-inflammatory status in these patients.

**Pregnant women with or without COVID-19 displayed a similar peripheral blood cell landscape distribution but different amounts of low-density granulocytes.** To assess whether a different peripheral blood landscape distribution was present in pregnant women with or without COVD-19, we used mass cytometry and a panel of 38 markers to define different subpopulations of T, B, NK, DC, $\gamma\delta$ cells, monocytes, and low-density neutrophils (LDN). The representation of different cell distribution of CD45-expressing leukocytes, visualized by the Uniform Manifold Approximation and Projection (UMAP) approach, is reported in Fig. 1A and supplementary Figs. 1–3. Unsupervised analysis revealed a total of 27 main clusters (Fig. 1B and Fig. 2); CD4+ and CD8+ T cells and B lymphocytes have been identified, and separately analyzed as described more in detail in the next paragraph.

The expression of CD16 and CD14 allows the identification of three types of monocytes, i.e., classical, intermediate, and non-classical. These cells are modified during pregnancy: the subpopulation of intermediate monocytes increases, while classical monocytes decrease and there are no changes in the nonclassical subpopulation[23]. Here, we show that all monocyte populations were not phenotypically different between infected or noninfected women, indicating that likely these cells do not participate in the creation of an inflammatory milieu.

CD16 and CD56 are largely used to classify NK cells, and in particular, three different clusters of NK can be recognized (early NK, mature NK expressing or not CD57). NK cells belong to the complex family of innate lymphoid cells that participate in tissue immunity. They are found in the placental decidua (dNK), are essential for successful fetal implantation, and are involved in tissue modifications and in forming new vessels[24]. They also interact with myelomonocytic cells to favor the development of regulatory T cells (Treg), that have a key role in immunosuppression and induction of tolerance to the fetus. dNK can derive from the expansion of single mature CD56^dim clones, from the recruitment and maturation of CD56^bright NK cells, or from the development of tissue-resident CD56^bright NK cells independently of the circulating compartment[25]. It is known that during pregnancy, the number and activity of peripheral blood NK cells decrease[26]. Even if we could not distinguish between dim or bright cells, in both infected or uninfected pregnant women, we found a significant decrease in circulating mature NK cells (i.e., those expressing CD57), indicating that SARS-CoV-2 was not able to alter the delicate equilibrium that regulates these highly reactive cells.

In the groups of pregnant women, we found similar proportions of lymphocytes expressing the $\gamma\delta$ T-cell receptor. In particular, three clusters of $\gamma\delta$ T cells were studied: naive cells expressing CD45RA plus CCR7, and effector memory (EM) expressing CD45RA (EMRA), expressing or not CD57. $\gamma\delta T$ cells play an active role in the tolerance of paternal antigens.

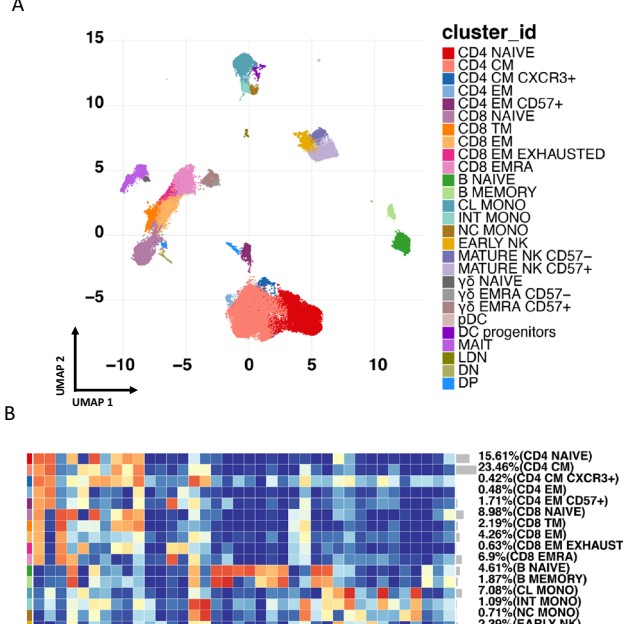

**Fig. 1 Unsupervised analysis of mononuclear cells in peripheral blood and their characterization. A** Uniform Manifold Approximation and Projection (UMAP) representation of PBMC landscape. Each color indicates a different cell population. CM, central memory; EM, effector memory; TM, transitional memory; CL, classical monocytes; INT, intermediate monocytes; NC, nonclassical monocytes; NK, natural killer cells; $\gamma\delta$, T cells expressing the $\gamma\delta$ T-cell receptor; DC, dendritic cells; MAIT, mucosal-associated invariant T cells; LDN, low-density neutrophils; DN, CD4-, CD8-T lymphocytes; DP, CD4+, CD8+ T lymphocytes. **B** Heatmap representing different clusters identified by FlowSOM, with relative identity and percentages in all groups of subjects. The color in the heatmap represents the median of the arcsinh, 0–1 transformed marker expression calculated over cells from all the samples, varying from blue for lower expression to red for higher expression.

They display a regulatory function, which supports tolerance toward the semiallogenic fetus, and exert a valid defense against pathogens[27]. It is to note that the frequency and activation status of $\gamma\delta$ T cells in patients with COVID-19 was found significantly lower than in matched healthy donors[28,29].

In pregnancy, proportions of myeloid dendritic cell (mDC) and plasmacytoid dendritic cell (pDC) decrease in the second trimester of pregnancy, but subsequently increase in late pregnancy becoming more activated, expressing increasing proportions of costimulatory markers, and producing inflammatory cytokines[30]. During COVID-19 infection, DC were significantly reduced with functional impairment, and the ratios of conventional DC to pDC were increased in severe patients. By evaluating the expression of CD123, we could identify two clusters of DC, expressing or not CD11c. pDC are CD11c-C123+ cells, while the CD11c+C123+ DCs represent DC progenitors[31]. Here, similar percentages of DC and their progenitors were found in pregnant women and healthy donors.

Mucosal-associated invariant T (MAIT) cells are antimicrobial T cells able to recognize bacterial metabolites and can function as innate-like sensors and mediators of antiviral responses, playing

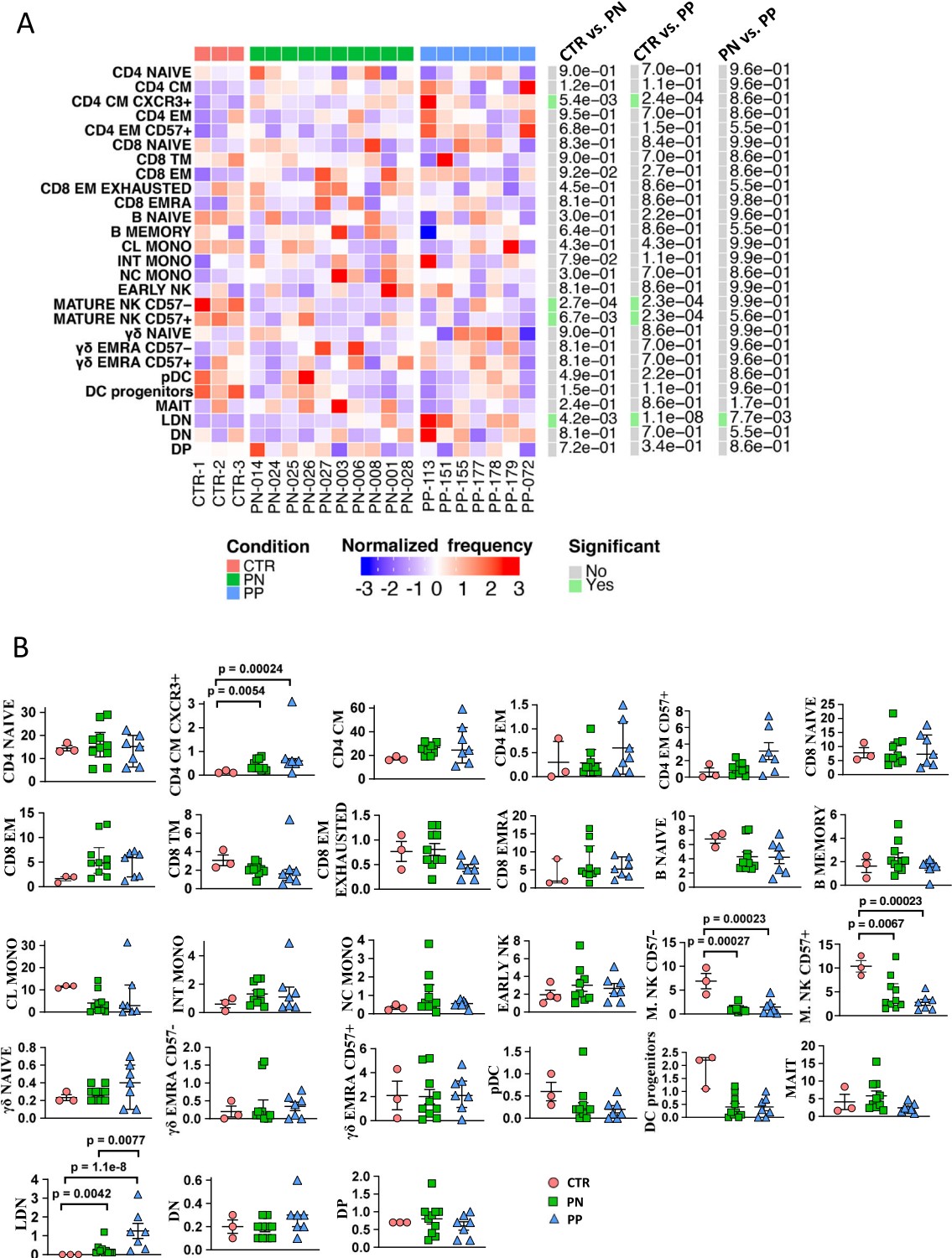

**Fig. 2 Detailed statistical analysis of PBMC landscape. A** Differential analysis performed using generalized linear mixed model (GLMM) between nonpregnant women (CTR; bar color: salmon; n = 3), uninfected pregnant women (PN; green, n = 10) and infected pregnant women (PP; azure, n = 7). The heat represents arcsine-square-root-transformed cell frequencies that were subsequently normalized per cluster (rows) to mean of zero and standard deviation of one. The color of the heat varies from blue indicating relative underrepresentation to red indicating relative overrepresentation. Bar and numbers at the right indicate significant differentially abundant clusters (green) and Bonferroni-adjusted p-values obtained from GLMM statistical test. **B** Analysis of the different cell populations identified as in panel **A**. A significant difference was observed in the percentage of low-density neutrophils (LDN), in infected pregnant women (PP) vs. uninfected pregnant (PN) or vs. healthy donors (CTR). The dot plots show the relative abundancies of 27 populations found within precleaned CD45+ live cells. Values (dots) for the three conditions are matched with the color used in **A**. Data represent individual percentage values (dots), median (center bar), and SEM (upper and lower bars). Generalized linear mixed model (GLMM) with Bonferroni correction has been used. The exact p-value is indicated in the figure. Source data are provided as a Source Data file.

also a role in the response to anticancer therapy[32]. MAIT cells accumulate at term pregnancy in the maternal blood that flows into the intervillous space inside the placenta, where they are recruited by chemotactic factors[33]. Patients with active COVID-19 infection displayed a decrease in circulating MAIT cell compartment that is characterized by strong activation[34]. Here, MAIT were defined by the expression of CD3, CD161, and the lack of expression of CD56, and similar percentages were present in the different groups analyzed.

There is a gradual but marked increase of blood levels of neutrophils during the first trimester of pregnancy. Neutrophil populations present in the low-density fraction after Percoll isolation, are a heterogeneous mixture of segmented mature neutrophils and morphologically immature (banded cells and ring-shaped nuclei) neutrophils [i.e., low-density neutrophils (LDN)]. When compared with high-density neutrophils, LDN possess decreased phagocytic activity, impaired reactive oxygen species (ROS) production, and a diminished capacity to inhibit CD8+ T-cell proliferation[35]. Typically, isolation of PBMC by Ficoll gradient removes neutrophils but not LDN. Thus, we could analyze LDN population on the basis of the expression of markers present in neutrophils, and LDN were expressing CD66b, CXCR1, CD11b, and CD16. We found increased levels of LDN in infected pregnant women, a phenomenon that has been described in COVID-19 patients and has been related to the severity of the disease. It has been hypothesized that LDN increase could be related to plasma levels of GM-CSF, whose elevation has been recently described in patients with COVID-19 pneumonia[19].

We investigated separately CD4+ and CD8+ T cells, and B lymphocytes (Figs. 3, 4 and 5, respectively). We could find 14 main clusters indicating different subpopulations of CD4+ T cells, 13 clusters of CD8+ T cells, and 6 clusters of B lymphocytes. In general, a few differences were present between healthy donors and pregnant women (either positive or negative), but women with or without SARS-CoV-2 infection were almost identical as far as all of the clusters of the T cells are concerned. Concerning B cells, pregnant women with or without infection were similar; both groups however displayed more plasmablasts, but less memory-switched cells than healthy donors.

Overall, we found that almost all analyzed clusters were not statistically different between pregnant women with and without COVID-19. Indeed, cells belonging to innate immunity (sub-population of monocytes, NK, DC) or adaptive (B and T cells) were represented in a similar percentage in pregnant women with or without COVID-19 infection.

**Expression of master-regulator genes and chemokine receptors on CD4+ and CD8+ T cells is similar in COVID-19 pregnant women and healthy controls.** To further characterize CD4+ and CD8+ T cells and better define the helper capability of T cells in terms of acting as Th1, Th2, or Th17, we investigated the expression of different chemokine receptors together with that of master-regulator genes such as CCR6, CD161, CXCR3, CCR4, GATA3 and TBET. Th1 was defined as CXCR3+TBET+, Th2 was defined as GATA3+, CCR4+, and Th17 was defined as CD161+CCR6+. The gating strategy for the identification of CD4+ and CD8+ T cells is reported in Supplementary Fig. 4.

Similar percentages of CD4+ T cells expressing CCR4, CCR6, CD161, CXCR3, TBET, and GATA3 were found among healthy donors and pregnant women with or without COVID-19. Moreover, no differences were found in terms of percentages of Th1, Th2, and Th17 among healthy donors and pregnant woman with or without COVID-19 (Fig. 6A). Concerning CD8+ T cells,

we found comparable percentages of cells expressing different master-regulator genes or chemokine receptors (Fig. 6B).

**T cells from COVID-19 pregnant women and healthy pregnant women are fully functional in terms of proliferation and cytokine production.** We investigated CD4+ and CD8+ T cells also from a functional point of view. Proliferative capability and cytokine production were evaluated and quantified after in vitro stimulation with anti-CD3/CD28. The gating strategies for studying the proliferation capacities and cytokine production are reported in Supplementary Figs. 5 and 6, respectively. By using the method of CFSE dilution for evaluating cell proliferation (Fig. 7A), we found that CD4+ T cells from pregnant women had a small but significant increase either in the proliferation index or in the division index, more marked in those who were not infected. No differences were found in CD8+ T cells (Fig. 6B).

We evaluated the amount of IL-4, TNF, IL-17A, and IFN-γ produced by CD4+ T cells and expression of CD107a (Fig. 8). In CD4+ T cells, we found no differences in terms of single-cytokine production (Fig. 8A). Similar results were obtained regarding CD8+ T cells (Fig. 8B), that gave comparable results in the three groups investigated.

**Correlogram reveals different correlation among several laboratory parameters.** We used all the data available in some pregnant women with or without the infection and built a cor-relation matrix, visualized in Fig. 9 and Supplementary Fig. 7. It is to note that in those who had the virus, but not in the others, highly significant positive correlations were present among the percentage of LDN (boxed in green in the first column) among mononuclear cells and plasma levels of molecules such as CCL2, CCL3, CCL4, CCL-5, CCL11, CCL19, CD40L, CXCL1, CXCL2, IL-1α, IL-1β, IL-2, IL-4, IL-7, IL-10, IL-13, IL-17E, IL-33, PDGF-AA, PDGF-AB/BB, EGF, IFN-α, and IFN-γ. Several positive correlations among plasma cytokines were then found, that were much more marked in SARS-CoV-2-infected pregnant women (Fig. 9 and Sup Data 2–4). The correlations between LDN and several soluble molecules, including inflammatory cytokines, that were present only in infected women deserve further attention.

## Discussion

In this study, we describe the main immunological features in pregnant women with paucisymptomatic or asymptomatic SARS-CoV-2 infection. To this regard, plasma level of 62 cytokines was measured together with the distribution of innate and adaptive immunity cells within PBMC. The main findings of our study are that, in comparison with control pregnant women, those with asymptomatic or mild COVID-19 are characterized by (i) dif-ferent plasma levels of a few pro-inflammatory and anti-inflammatory cytokines; (ii) lack of high plasma levels of IL-6; (iii) lower levels of D-dimer; (iv) similar distributions of different populations of PBMC; (v) significantly higher amounts of circu-lating low density neutrophils. Moreover, T- and B-cell subsets were in-depth characterized, and we found that these cells were also able to maintain their functional properties, in terms of cell proliferation and cytokine production.

From a clinical point of view, the majority of pregnant women affected by COVID-19 are asymptomatic or paucisymptomatic. In them, immune responses and changes have never been studied in detail, and scanty data describe immunological changes due to SARS-CoV-2 infection. Most studies reported that pregnant women posi-tive for COVID-19 do not experience severe symptoms and gave birth to healthy babies, so the infection seems to impact neither the pregnancy nor the newborns[10], even in a case where a pregnant

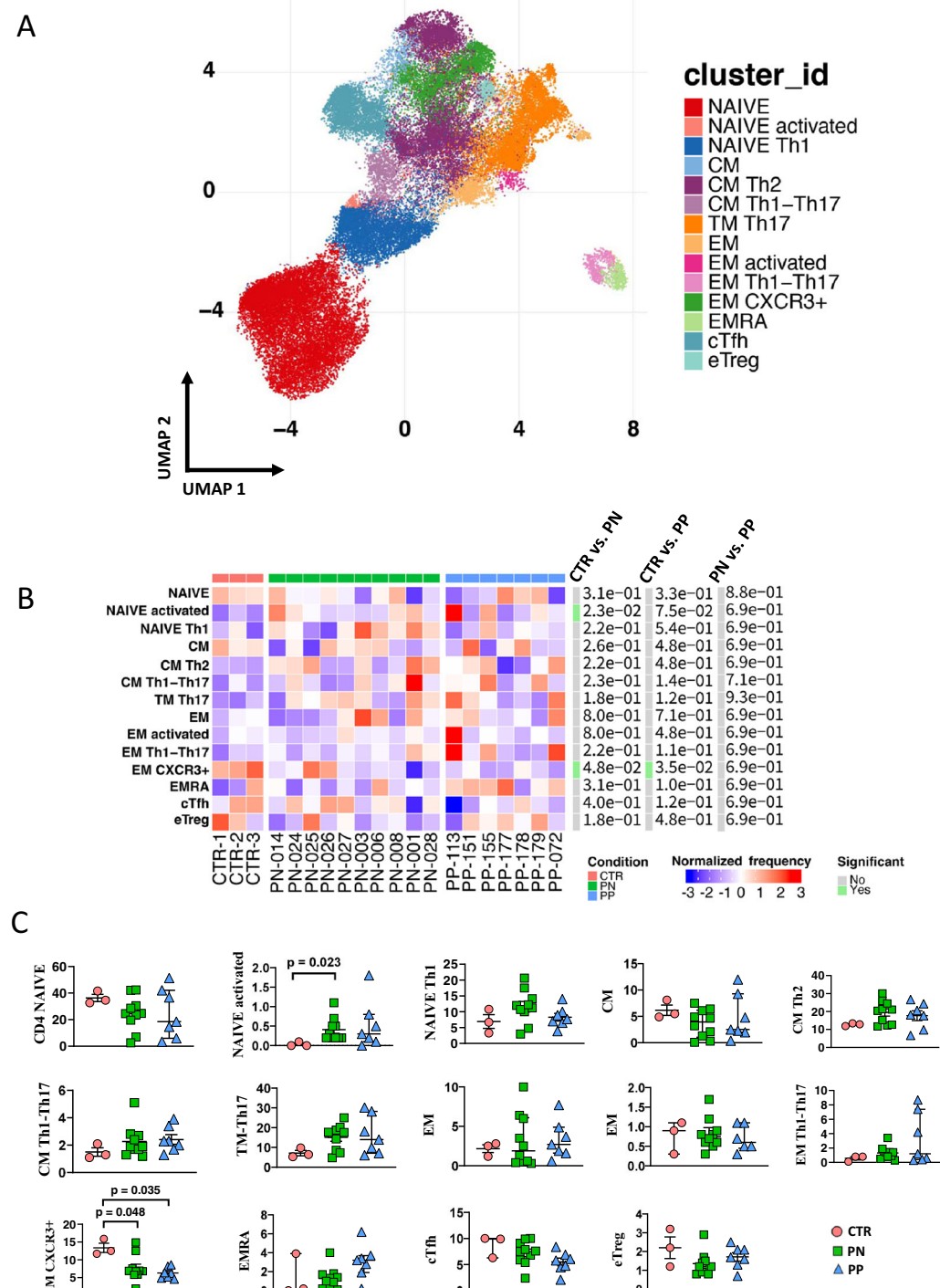

**Fig. 3 Reclustering of CD4+ T cells reveals similar distribution of cells between infected or noninfected pregnant women and donors. A** Uniform manifold approximation and projection (UMAP) representation of CD4 T-cell landscape. Each color indicates one of the 14 different clusters. CM central memory, TM transitional memory, EM effector memory, EMRA terminally differentiated T cells, cTfh circulating follicular T helper cells, eTREG effector-regulatory T cells. **B** Differential analysis performed using generalized linear mixed model (GLMM) between nonpregnant women (CTR; bar color: salmon; *n* = 3), uninfected pregnant women (PN; green, *n* = 10), and infected pregnant women (PP; azure, *n* = 7). The heat represents arcsine-square-root-transformed cell frequencies that were subsequently normalized per cluster (rows) to mean of zero and standard deviation of one. The color of the heat varies from blue indicating relative underrepresentation to red indicating relative overrepresentation. Bar and numbers at the right indicate significant differentially abundant clusters (green) and adjusted *p* values. **C** Statistical analysis of the different cell populations identified as in panel **B**. Comparison among infected pregnant women (PP) vs. uninfected pregnant (PN) or vs. healthy donors (CTR). The dot plots show the relative abundances of 14 populations found within CD4+ T cells. Values (dots) for the three conditions are matched with the color used in **B**. Data represent individual percentage values (dots), median (center bar), and SEM (upper and lower bars). Generalized linear mixed model (GLMM) with Bonferroni correction has been used. The exact *p*-value is indicated in the figure. Source data are provided as a Source Data file.

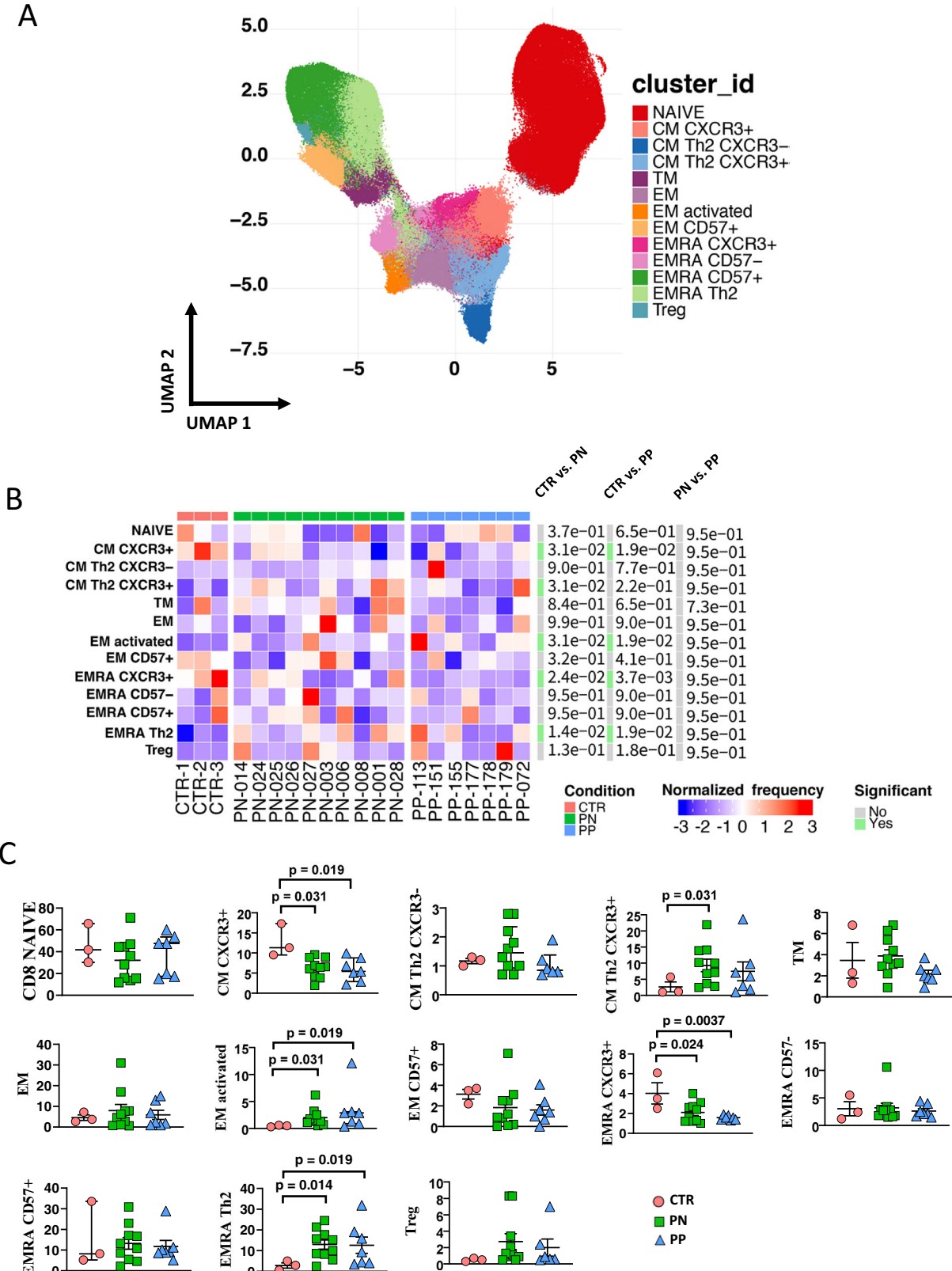

woman with critical COVID-19 has been described[36]. Both the mother and son showed lymphopenia, along with a delayed immunoglobulin (Ig) response due to a low number of Ig-switched B cells and a very small compartment of naive T and B cells. In contrast with recent data showing that COVID patients display profound derangement in the B-cell compartment[37], we found that the number

and quality of B cells were similar in infected or uninfected women, who both showed small but significant changes in memory B cells and in plasmablasts.

SARS-CoV-2 infection leads to a process called "cytokine storm", where a variety of cytokines, from those proinflammatory, to anti-inflammatory, or those indicating a skewing

**Fig. 4 Reclustering of CD8+ T cells reveals similar distribution of cells between infected or noninfected pregnant women and donors. A**. Left panel: Uniform manifold approximation and projection (UMAP) representation of CD8 T-cell landscape. Each color indicates one of the 13 different clusters. CM central memory, TM transitional memory, EM effector memory, EMRA terminally differentiated T cells, TREG regulatory T cells. **B** Differential analysis performed using generalized linear mixed model (GLMM) between nonpregnant women (CTR; bar color: salmon; $n = 3$), uninfected pregnant women (PN; green, $n = 10$), and infected pregnant women (PP; azure, $n = 7$). The heat represents arcsine-square-root-transformed cell frequencies that were subsequently normalized per cluster (rows) to mean of zero and standard deviation of one. The color of the heat varies from blue indicating relative underrepresentation to red indicating relative overrepresentation. Bar and numbers at the right indicate significant differentially abundant clusters (green) and adjusted p values. **C** Statistical analysis of the different cell populations identified as in panel **A**. Comparison among infected pregnant women (PP) vs. uninfected pregnant (PN) or vs. healthy, nonpregnant donors (CTR). The dot plots show the relative abundancies of 13 populations found within CD8+ T cells. Values (dots) for the three conditions are matched with the color used in **B**. Data represent individual percentage values (dots), median (center bar), and SEM (upper and lower bars). Generalized linear mixed model (GLMM) with Bonferroni correction has been used. The exact p-value is indicated in the figure. Source data are provided as a Source Data file.

toward Th1, Th2, or Th17, are produced[37,38]. The same process was already described during sepsis, where this overwhelming phenomenon may lead to multiple-organ failure. Few data, if any, exist that describe in detail plasma cytokine levels, nor that report cytokine production in lymphocytes from pregnant women with COVID-19. Here we found increased level of different soluble molecules, such as pro- and anti-inflammatory cytokines. During the cytokine storm, pro-inflammatory molecules are first produced to contrast the infection, but anti-inflammatory molecules are needed to dampen inflammation. We reported that patients affected by COVID-19 showed an increase of plasma level of cytokines with different properties, such as CCL4, IL-10, and PD-L1[39]. The same phenomenon has been shown in sepsis where there is the concomitant production of IL-1RA and IL-10[40] and high level of both APRIL and BAFF[41].

The correlations here described between different cytokines or between a given cytokine and hematochemical and immunological parameters are quite complex. It is interesting to note that in infected women, but not in pregnant-negative women, LDN levels show significant correlations with a number of cytokines and soluble molecules that have different functions and exert different effects on these cells. For example, several functional activities of polymorphonuclear neutrophils are subject to regulation by immunosuppressive molecules like IL-4, IL10, and IL-13, but also to the modulation by molecules favoring their proinflammatory capacity such as IL-1, IL-17, IL-18, and different interferons. It has been shown that two major populations of LDN exist, such as immunosuppressive LDN, also known as polymorphonuclear myeloid-derived suppressor cells that are typically found in cancer, pregnancy, infections, and systemic inflammation[42–44], and proinflammatory LDN, or low-density granulocytes, that are typically found in autoimmune diseases. At present, standardized markers are not yet available to define such populations, and their pro- or anti-inflammatory function can be only verified by testing in vitro their activity[45]. On the one side, considering that several cytokines quoted here can also be produced by cells of the granulocytic family, the correlations of LDN with cytokines with different activity could be due to the existence of the two cell populations described above. On the other, even if we could not measure any functional parameter, it can be hypothesized that, in our cohort of pregnant women infected by SARS-CoV-2, LDN with immunosuppressive capabilities were present, and were able to control inflammation.

During pregnancy, the immune system goes through some modifications to create the proper environment for fetal growth and for immunological tolerance. Any inflammatory milieu that triggers a Th1 immune response can clearly be dangerous for the fetus, and the anti-inflammatory immunological shift that occurs in pregnancy can partially be explained by alterations in hormonal levels, including progesterone, estradiol, and other proteins like leukemic-inhibitory factor, as well as prostaglandins[46,47]. This can

well explain the low levels of cytokines that regulate T-cell activities like IL-2, IL-4, and IL-5, that we have found in both groups of pregnant women. Similarly, the balance between pro- and anti-inflammatory cytokines goes in the direction of the latter ones, as shown, for example, by the opposite directions taken by IL-1, IL-33, and IL-1RA.

Despite changes in the plasmatic levels of a variety of cytokines also produced by PBMC, we found similar distribution of different PBMC populations between pregnant women with and without infection. Therefore, it could be hypothesized that an equilibrium in the cytokine network is reached that in pregnant women with COVID-19 does not cause damages to the immune system and is probably able to balance the opposite biological effects of several cytokines, creating or even reinforcing immune tolerance.

Interestingly, it is likely that the changes described here not only do not alter the whole immune response to the virus, but also do not cause any significant change in the asset of circulating lymphocytes and monocytes. However, an increase of LDN "infiltrating" the PBMC population was found both in pregnant-positive and pregnant-negative women. Due to pregnancy, immature forms of neutrophils can be detected in the circulation. As described above, LDN display an anti-inflammatory capacity mediated by the expression of arginase that causes a depletion of L-arginine and consequently the downregulation of lymphocyte responses. Indeed, they can efficiently suppress CD4+ and CD8+ T-cell proliferation and can also skew the balance between Th1 and Th2 lymphocytes toward a Th2/ regulatory T-cell phenotype. This could be linked to the immune suppressive environment observed within pregnant women[48,49].

We are well aware of the limitations of this study. First of all, the relatively low number of patients studied. Second, we could only investigate a cohort of asymptomatic or mildly symptomatic pregnant women. Few cases of pregnant women with severe COVID-19 are described, but no data exist regarding a deep immune characterization of those patients, nor on the presence of predictive markers. Furthermore, data on hormone production would be of interest to better investigate its importance in this situation. Despite the limitations pointed out, this is likely the first detailed immunological study conducted on pregnant women with SARS-CoV-2 infection, and it shows that asymptomatic or paucisymptomatic infected pregnant women are characterized by a few changes in plasma cytokine levels, that cannot be compared with the classical, well-known cytokine storm that has been described during severe COVID-19.

Finally, we underline that in our cohort, SARS-CoV-2 had no impact on newborns. None was infected nor suffered of any disturbance after birth. This is in agreement with a recent study showing no virus in maternal or cord blood (despite detection in the women's respiratory system), no signs of the virus in placenta, and no evidence of viral transmission to newborns[11]. Thus, it can be hypothesized that infection of the fetus does not occur because

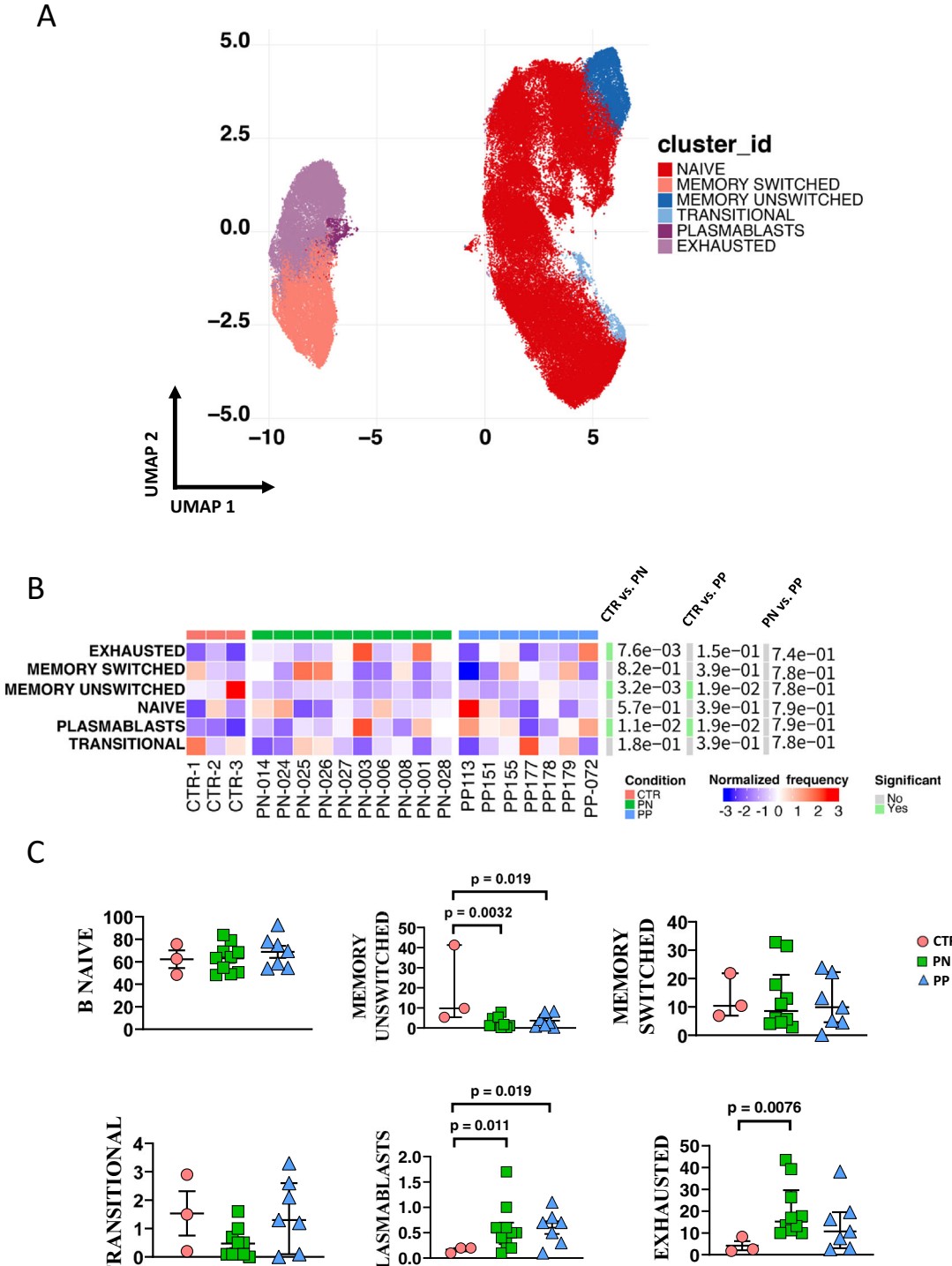

**Fig. 5 Reclustering of CD19$^+$ B cells reveals similar distribution of cells between infected or noninfected pregnant women and donors. A**. Left panel: Uniform manifold approximation and projection (UMAP) representation of B-cell landscape. Each color indicates one of the six different clusters. **B** Differential analysis performed using generalized linear mixed model (GLMM) between nonpregnant women (CTR; bar color: salmon; $n = 3$), uninfected pregnant women (PN; green, $n = 10$), and infected pregnant women (PP; azure, $n = 7$). The heat represents arcsine-square-root-transformed cell frequencies that were subsequently normalized per cluster (rows) to mean of zero and standard deviation of one. The color of the heat varies from blue indicating relative underrepresentation to red indicating relative overrepresentation. Bar and numbers at the right indicate significant differentially abundant clusters (green) and adjusted $p$ values. **C** Detailed statistical analysis of the different cell populations identified as in panel **A**. Comparison among infected pregnant women (PP) vs. uninfected pregnant (PN) or vs. healthy donors (CTR). The dot plots show the relative abundancies of six populations found within B cells. Values (dots) for the three conditions are matched with the color used in **B**. Data represent individual percentage values (dots), median (center bar), and SEM (upper and lower bars). Generalized linear mixed model (GLMM) with Bonferroni correction has been used. The exact $p$-value is indicated in the figure. Source data are provided as a Source Data file.

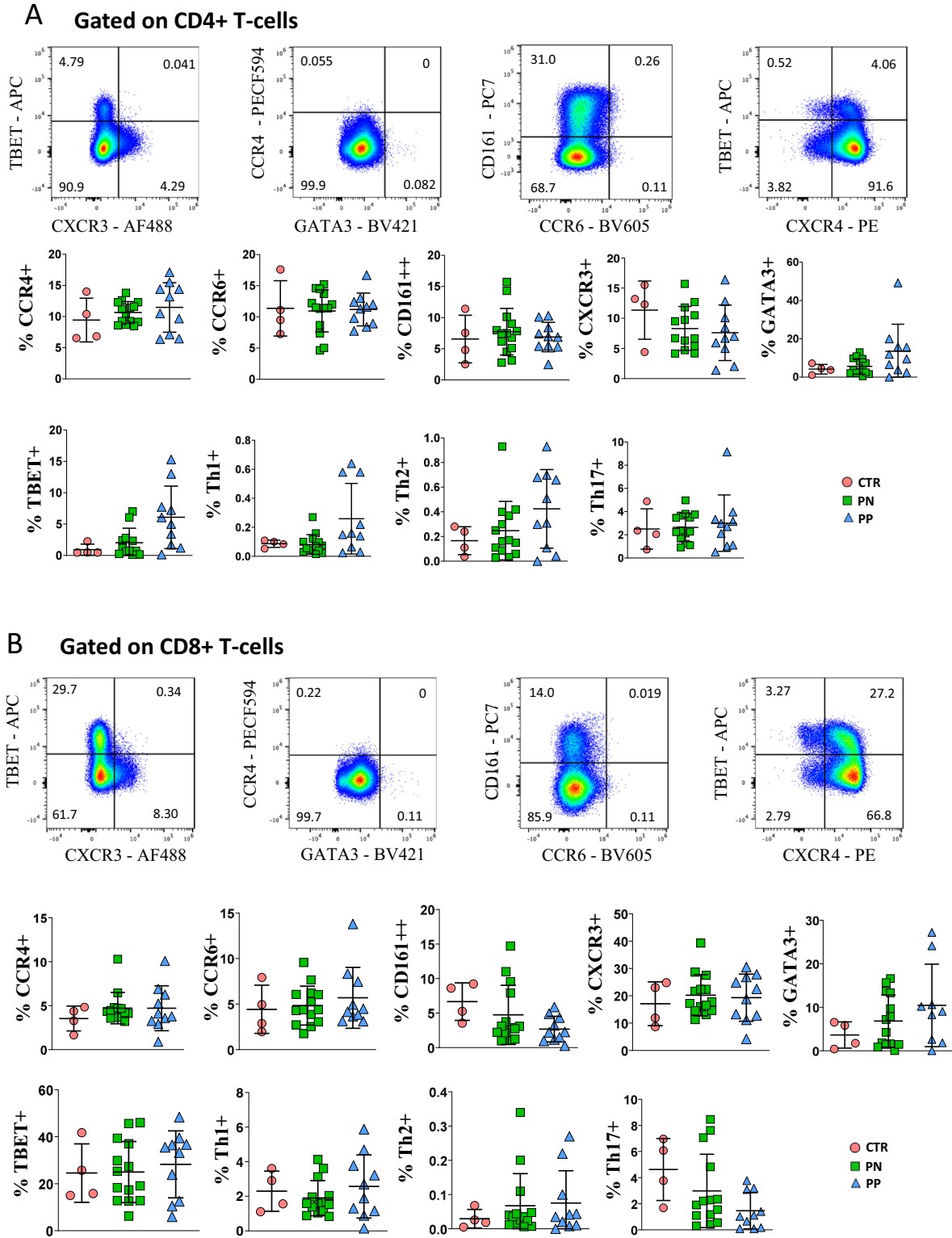

**Fig. 6 Phenotypic characterization of CD4 and CD8 T cells of COVID-19 pregnant women. A** Expression of different chemokine receptors and lineage-specifying transcription factors in gated CD4[+] T cells from healthy donors (CTR, $n = 4$), pregnant negative (PN, $n = 13$) and pregnant positive (PP, $n = 10$). Values (dots) for the three conditions are matched with the color used in Figs. 1–5. Data represent individual percentage values (dots), mean (center bar), and SD (upper and lower bars). Two-tailed Mann–Whitney $t$-test with Bonferroni correction has been used. Exact $p$-value is indicated in the figure, if significant. **B** Expression of different chemokine receptors and lineage-specifying transcription factors in gated CD8[+] T cells from healthy donors (CTR, $n = 4$), pregnant negative (PN, $n = 13$) and pregnant positive (PP, $n = 10$). Values (dots) for the three conditions are matched with the color used in Figs. 1–5. Data represent individual percentage values (dots), mean (center bar) and SD (upper and lower bars). Two-tailed Mann–Whitney $U$-test with Bonferroni correction has been used. The exact $p$-value is indicated in the figure, if significant.

## A   CD4 T cells

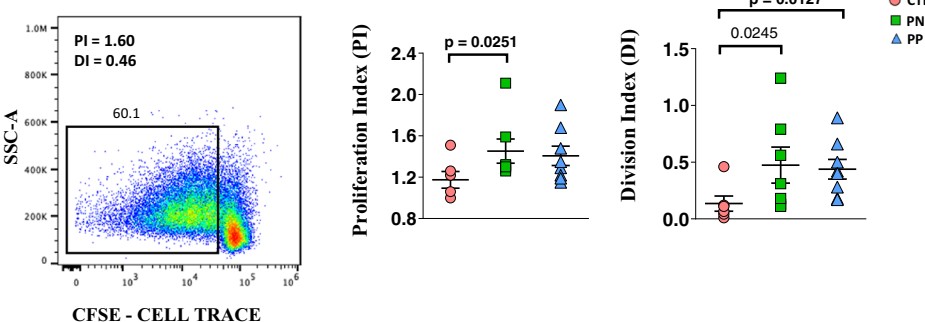

## B   CD8 T cells

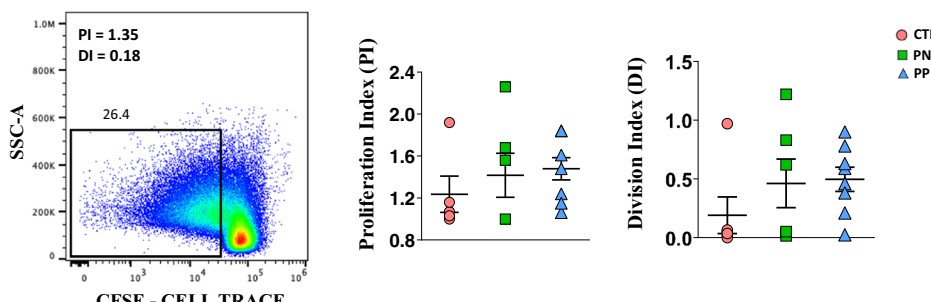

**Fig. 7 Proliferation of CD4 and CD8 T cells of COVID-19 pregnant women. A**. Proliferation index (PI) and division index (DI) in all CD4$^+$ T cells from the three groups studied. Data represent individual values (dots) from healthy donors (CTR, $n = 5$), pregnant negative (PN, $n = 6$), and pregnant positive (PP, $n = 7$). Values (dots) for the three conditions are matched with the color used in Figs. 1–5. Data represent individual percentage values (dots), mean (center bar) and SEM (upper and lower bars). Two-tailed Mann–Whitney $t$-test with Bonferroni correction has been used. The exact $p$-value is indicated in the figure, if significant. **B**. Proliferation index (PI) and division index in all CD8$^+$ T cells from the three groups studied. Data represent individual values (dots) from CTR ($n = 5$), pregnant negative (PN, $n = 6$), and pregnant positive (PP, $n = 7$). Values (dots) for the three conditions are matched with the color used in Figs. 1–5. Data represent individual percentage values (dots), mean (center bar), and SEM (upper and lower bars). Two-tailed Mann–Whitney $U$-test with Bonferroni correction has been used. The exact $p$-value is indicated in the figure, if significant.

of the reduced coexpression and colocalization of placental angiotensin-converting enzyme 2 and transmembrane serine protease 2, and that the immunosuppression present during pregnancy could protect not only most women from the cytokine storm, massive immune activation, and hyper inflammation, but also the fetus from a possible maternal immune aggression. In conclusion, our data might be useful for clinicians, as if the conditions of patients are stable, and infection remains mild or asymptomatic, there could be no risk of pregnancy complications neither for the mother nor for the newborn, so that in the large majority of the cases, an anticipated delivery should not be required.

## Methods

**Study design**. This is a case–control, cross-sectional, single-center study, approved by the local Ethical Committee (Comitato Etico dell'Area Vasta Emilia Nord, protocol number 177/2020, March 11th, 2020) and by the University Hospital Committee (Direzione Sanitaria dell'Azienda Ospedaliero-Universitaria di Modena, protocol number 7531, March 11th, 2020). Each participant, including healthy controls, provided informed consent according to Helsinki Declaration, and all uses of human material have been approved by the same committees. A total of 14 pregnant women with SARS-CoV-2 infection were included in the study; they had a median age of 33.8 years (range 19–39). Patients were matched for age and gender with 28 pregnant women negative to nasopharyngeal swab (median 33.9 years, range 18–42) and a total of 15 nonpregnant healthy women (CTR), median age 39 years (range 25–50 years). We recorded demographic data, medical history,

symptoms, signs, temperature, and the main laboratory findings from each patient. For details, see Sup Data 1.

Pregnant women were eligible for inclusion if they were aged 18 years or older, able to provide informed consent, and diagnosed with SARS-CoV-2 infection. Confirmed SARS-CoV-2 infection was defined as nasopharyngeal swab reverse transcription–polymerase chain reaction (RT-PCR) test results positive for SARS-CoV-2. According to routine methods, we could measure anti-SARS-CoV-2 IgM and IgG in 12 women positive to the swab, and nine had detectable plasma levels of IgM and IgG, while three were negative to both antibodies. All the 15 control women that we could test were negative.

The total number and type of leukocytes in peripheral blood was analyzed by a hemocytometer in the Clinical Laboratory of the University Hospital, that also analyze all the biochemical parameters quoted in the paper, according to routine methods.

**Blood collection**. Blood samples (up to 20 mL) were obtained after informed consent. In some donors, blood was obtained after diagnosis of SARS-CoV-2 infection. Peripheral blood mononuclear cells (PBMC) were isolated according to standard procedures and stored in liquid nitrogen until use. Plasma was collected and stored at −80 °C until use. Measurements were taken from individual patients; in the case of plasma, each measurement was performed in duplicate and only the mean was considered and shown.

**Mass cytometry analysis**. Thawed PBMC were washed twice with PBS and stained with Maxpar® Direct™ Immune Profiling Assay™ (Fluidigm), a dry 30-marker antibody panel (viability marker Cell-ID™ Intercalator-103Rh included) plus the addition of six drop-in catalog antibodies (Fluidigm), and two custom-conjugated mAbs, for a total of 38 markers. The markers were the following: CD3,

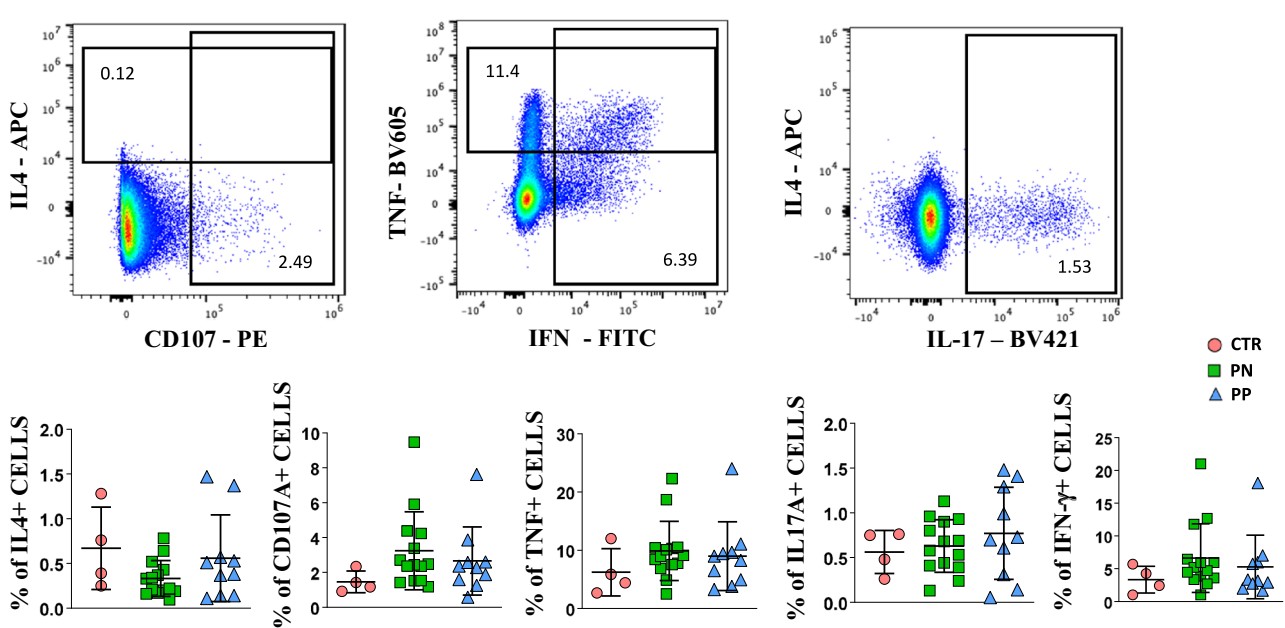

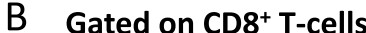

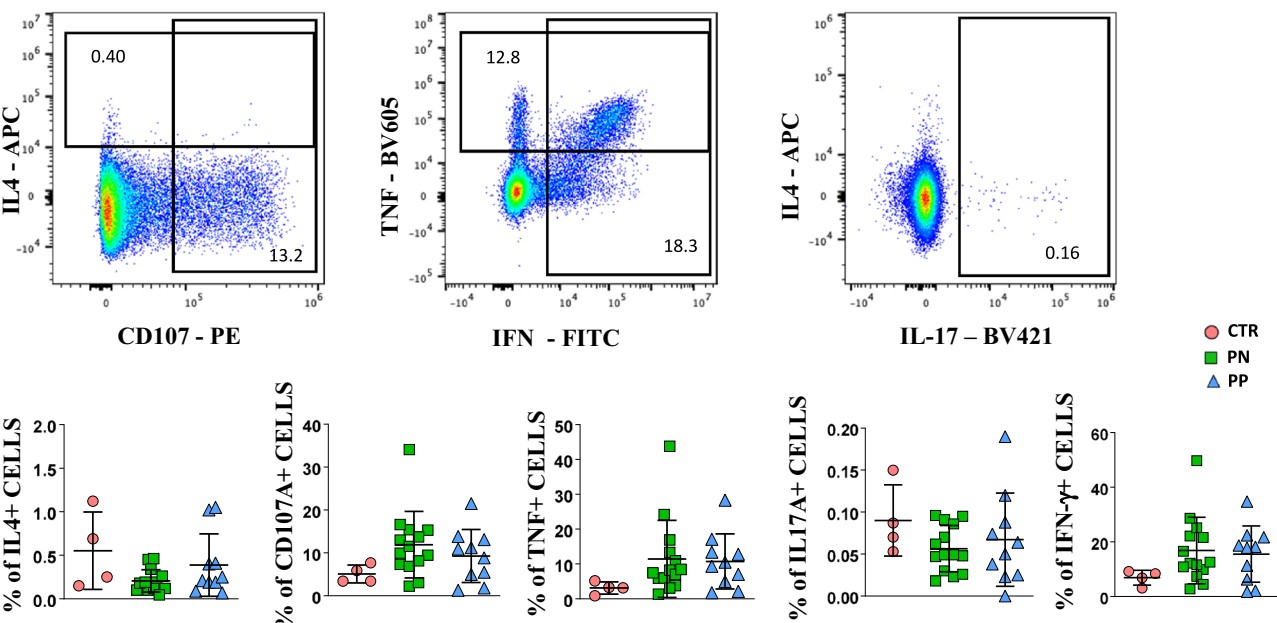

**Fig. 8 Functional characterization of CD4 and CD8 T cells of COVID-19 pregnant women. A** Comparison among the three groups of persons under investigation regarding the production of cytokines by CD4+ T cells after in vitro stimulation with anti-CD3/CD28. Data represent individual values from healthy donors (CTR, $n = 4$), pregnant negative (PN, $n = 14$), and pregnant positive (PP, $n = 10$), mean (center bar) ± standard deviation (upper and lower bars). Values (dots) for the three conditions are matched with the color used in Figs. 1–5. Data represent individual percentage values (dots), mean (center bar), and standard deviation (upper and lower bars). Two-tailed Mann–Whitney $t$ test with Bonferroni correction has been used. The exact $p$ value is indicated in the figure, if significant. **B** Comparison among the three groups of persons under investigation regarding the production of cytokines by CD8+ T cells after in vitro stimulation with anti-CD3/CD28. Data represent individual values from healthy donors (CTR, $n = 4$), pregnant negative (PN, $n = 14$), and pregnant positive (PP, $n = 10$), mean (center bar) ± standard deviation (upper and lower bars). Values (dots) for the three conditions are matched with the color used in Figs. 1–5. Data represent individual percentage values (dots), mean (center bar), and SD (upper and lower bars). Two-tailed Mann–Whitney $U$-test with Bonferroni correction has been used. The exact $p$-value is indicated in the figure, if significant.

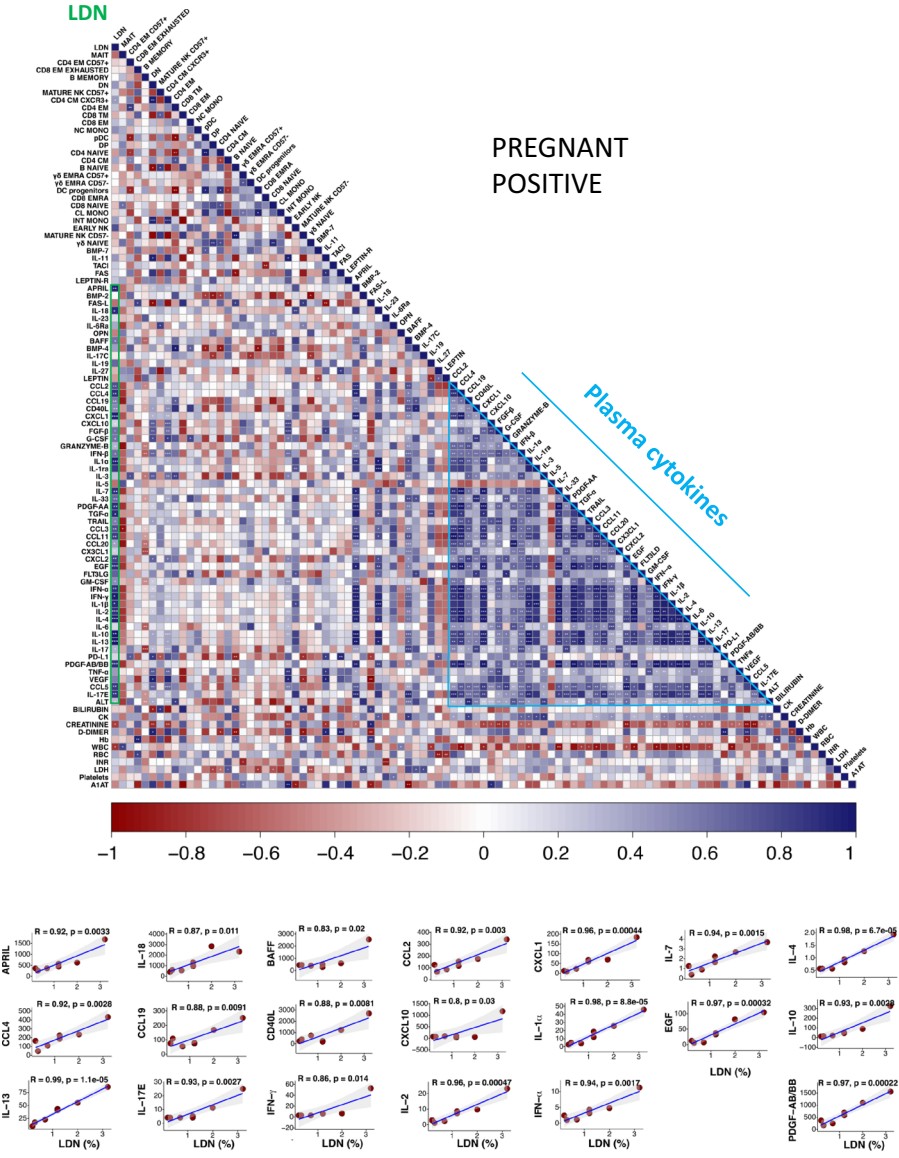

**Fig. 9 Correlogram showing the role of LDN in infected women, and the correlations among plasma cytokines.** Correlogram of pregnant positive women (PP). Spearman R ($\rho$) values are shown from red (−1.0) to blue (1.0); specific R values are indicated by the square color. Blank fields with dots indicate lack of signal. Spearman rank two-tailed *p*-value was indicated by *$P < 0.05$, **$P < 0.01$, and ***$P < 0.001$. Additional XY scatter plots, that specifically show the relationship between the variables that are most correlated with LDN, are displayed. Each scatter plot reports the regression line (blue), the Spearman R ($\rho$) value, the exact two-tailed *p*-value and the 95% confidence bands (light gray). All cytokines value are referred as pg/ml. PDGF-AB/BB refers to the ratio between the forms AA and AB of the platelet-derived growth factor (PDGF).

CD19, CD45, CD4, CD20, CD45RA, CD8, CD25, CD45RO, CD11c, CD27, CD56, CD14, CD28, CD57, CD16, CD38, CD66b, CCR7, CXCR3, CXCR5, HLA-DR,IgD, TCRγδ, CD123, CD127, CD161, CD294, CCR4, CCR6, CXCR1, PDL1, CD80, CD40, CD24, PD1-1, CD11b/MAC, CD21, and IgM. See Sup Data 5 for the complete list of mAbs used. At least 300,000 events were acquired per sample.

Data in FCS file format were normalized for intrafile and interfile signal drift using the FCS Processing tab in the CyTOF Software 6.7. The method is a two-step algorithm that first identifies the EQ Four Element Calibration Beads and then applies the dual-count values registered by the beads to calculate the normalization factor to be applied to the data.

**Representation of high-parameter flow cytometry.** Compensated and normalized Flow Cytometry Standard (FCS) 3.0 files were imported into FlowJo software version 10 (Becton Dickinson, San Josè, CA) and preprocessed excluding EQ Four Element Calibration Beads and doublets using Gaussian Discrimination parameters. Then, were selected live undamaged CD45+ and excluded artifact cells (CD3+CD19 +, or CD3+CD14+). All living CD45+ were exported for further analysis in R using Bioconductor libraries CATALYST (version 1.12.2)[50] and diffcyt (version 1.8.8)[51].

The data were transformed using arcsinh with *cofactor* = 5 to make the distributions more symmetric and to map them to a comparable range of expression. The main cell population identification was performed through unsupervised clustering using the FlowSOM (version 1.22.0) algorithm (K = 30). 2D visual representation was performed applying Uniform Manifold Approximation and Projection (UMAP). Then, the clusters identified as CD4+, CD8+, or CD19+ lymphocytes, were selected and reclustered separately to describe more in-depth the cellular distribution of each subpopulation. We used K = 15 for CD4+ and CD8+ T cells, while K = 9 for CD19+ cells. Clusters with similar marker distribution were merged. Then we reapplied UMAP for dimensionality reduction and visualization purposes. Statistical analysis was performed using generalized linear mixed models (GLMM) applying as FDR cutoff =0.05. Of note, for each step of clustering performed, clusters with similar marker distribution were manually merged.

**Polychromatic flow cytometry**

*T-cell characterization.* For the analysis of T cells skewing toward Th1, Th2, or Th17, and chemokine-receptor expression, thawed PBMC were washed twice with PBS and stained with the viability marker AQUA LIVE DEAD (ThermoFisher).

Then, up to 1 million cells were washed and stained at 37 °C with the following mAbs: anti-CXCR3-AF488, -CXCR4-PE. Cells were washed again and stained at room temperature with anti-CD161-PC7, -CCR6-BV605, -CCR4-PE-CF594, -CD4-AF700, and -CD8- APC-Cy7. Cells were washed, fixed, and permeabilized using Foxp3/Transcription Factor Staining Buffer Set (ThermoFisher). Finally, cells were stained with anti-GATA3-BV421 and anti-TBET-APC and washed. A minimum of 500,000 PBMC were acquired by using Attune NxT acoustic focusing flow cytometer (ThermoFisher). FCS data were acquired in list mode by Attune nxT Software v4.2. mAbs used are listed in Sup Data 6.

*Proliferation assay*. Cells were stimulated for six days in resting conditions, or after stimulation with anti-CD3 plus anti-CD28 mAbs (1 µg/mL each, Miltenyi Biotech, Bergisch Gladbach, Germany) and with 20 ng/mL IL-2. The fluorescent dye 5,6-car-boxyfluorescein diacetate succinimidyl ester was used at a concentration of 1 µg/mL (ThermoFisher) according to standard procedures[52]. Flow cytometric analyses for the identification of cycling cells belonging to different T-cell populations were performed by gating CD4[+], CD8[+] T cells, and CD19+ B cells. mAbs used are listed in Sup Data 6.

*In vitro stimulation and intracellular cytokine staining*. For functional assays on cytokine production by T cells, thawed isolated PBMCs were stimulated for 16 h at 37 °C in a 5% CO2 atmosphere with anti-CD3/CD28 (1 µg/mL) in complete culture medium (RPMI 1640 supplemented with 10% fetal bovine serum and 1% each of L-glutamine, sodium pyruvate, nonessential amino acids, antibiotics, 0.1 M HEPES, and 55 µM β-mercaptoethanol). For each sample, at least 2 million cells were left unstimulated as negative control, and 2 million cells were stimulated. All samples were incubated with a protein-transport inhibitor containing brefeldin A (Golgi Plug, Becton Dickinson) and previously titrated concentration of CD107a-PE. After stimulation, cells were stained with LIVE-DEAD Aqua (ThermoFisher Scientific) and surface mAbs recognizing CD4 AF700, and CD8 APC-Cy7 (Biolegend, San Diego, CA, USA). Cells were washed with stain buffer and fixed and permeabilized with the cytofix/cytoperm buffer set (Becton Dickinson) for cytokine detection. Cells were next stained with previously titrated mAbs recognizing CD3 PE-Cy5, IL-17 BV421, TNF BV605, IFN-γ FITC, IL-4 APC, or granzyme-B BV421 (all mAbs from Biolegend). Then, a minimum of 100,000 cells per sample were acquired on a Attune NxT acoustic cytometer (ThermoFisher)[53]. FCS data were acquired in list mode by Attune nxT Software v4.2. mAbs used are listed in Sup Data 6.

*Quantification of cytokine plasma levels*. The plasma levels of 62 molecular species were quantified using a Luminex platform (Human Cytokine Discovery, R&D System, Minneapolis, MN) for the simultaneous detection of the following molecules: G-CSF, PDGF-AA, EGF, PDGF-AB/BB, VEGF, GM-CSF, FGF, GRZB, IL-1A, IL-1RA, IL-2, IL-27, IL-4, IL-6, IL-10, IL-13, TNF, IL-17C, IL-11, IL-18, IL-23, IL-6RA, IL-19, IFN-B, IL-3, IL-5, IL-7, IL-12p70, IL-15, IL-33, TGF-B, IFN-G, IL-1B, IL-17, IL-17E, CCL3, CCL11, CCL20, CXCCL1, CXCL2, CCL5, CCL2, CCL4, CCL19, CXCL1, CXCL10, PD-L1, FLT-3, TACI, FAS, LEPTIN R, APRIL, OPN, BAFF, LEPTIN, BMP4, CD40 LIGAND, FAS LIGAND, BMP7, BMP2, and TRAIL, according to the manufacturer's instruction.

*Analysis of the correlations among all parameters*. To identify possible correlations among the parameters we have studied, we have designed a table containing (i) all 27 cluster percentages obtained using the unsupervised analysis on living CD45+ cells; (ii) 12 biochemical parameters reported in Sup Data 2; (iii) 61 plasma cytokines out of 62 because all values of IL-17p70 were identical. Pairwise correlations between variables were calculated and visualized as a correlogram using R function corrplot v0.84. Spearman's rank-correlation coefficient ($\rho$) was indicated by color scale; two-tailed *p*-value calculated using *cor.mtest* function of corrplot v0.84 was indicated by *$P < 0.05$, **$P < 0.01$, and ***$P < 0.001$. All variables were displayed using original order without applying any hierarchical clustering. Single XY scatter plot drew using *ggscatter* function of ggpubr v0.4.0 package, reporting rho ($\rho$) and exact two-tailed *p*-value.

*Statistical analysis*. High-dimensional cytometric analysis was performed by using differential discovery in high-dimensional cytometry via high-resolution clustering, while the statistical analysis was performed using generalized linear mixed models (GLMM) applying as FDR cutoff = 0.05. Quantitative variables were compared using Mann–Whitney *U*-test. Data are represented as individual values, means, and standard errors of the mean. Statistical analyses were performed using Prism 6.0 (GraphPad Software Inc., La Jolla, USA).

A table containing all the percentages of the 27 clusters identified using the unsupervised analysis on living CD45+ cells, 12 clinical parameters and 62 plasma cytokines, was created for correlation analysis (Fig. 9, and Sup Data 2–4).

**Reporting summary**. Further information on research design is available in the Nature Research Reporting Summary linked to this article.

## Data availability
The flow cytometry data generated in this study have been deposited in the flowrepository. org database under accession code FR-FCM-Z3GH [https://flowrepository.org/experiments/ 3601]. The raw data generated in this study are provided in the Source Data file. Source data are provided with this paper.

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

## Acknowledgements
S.D.B. and L.G. are Marylou Ingram Scholar of the International Society for Advancement of Cytometry (ISAC) for the period 2015–2020 and 2020–2025, respectively. We gratefully acknowledge Fluidigm Corporation (San Francisco, CA) for generous and unconditional support. Drs. Paola Paglia (ThermoFisher Scientific, Monza, Italy), Leonardo Beretta (Beckman Coulter, Milan, Italy), and Paolo Santino (Fluidigm Co., San Francisco, CA) are acknowledged for their support in providing reagents and materials in this study, and for precious technical suggestions. This study also received unrestricted donations from Glem Gas spa (San Cesario, Modena, Italy), Sanfelice 1893 Banca Popolare (San Felice S.P., Modena, Italy) and Rotary Club Distretto 2072 (Clubs: Modena, Modena L.A. Muratori, Carpi, Sassuolo, and Castelvetro di Modena), C.O.F.I. M. spa & Gianni Gibellini, Franco Appari, Andrea Lucchi, Federica Vagnarelli, Biogas Europa Service & Massimo Faccia, Pierangelo Bertoli Fans Club, and Alberto Bertoli, Maria Santoro, Valentina Spezzani and BPER Banca. Finally, special thanks to the patients who donated blood to participate in this study.

## Author contributions
S.D.B., L.G., D.L.T., A.P., A.Q., C.P., G.A., S.D., D.L., J.N., R.B., L.F., A.I., and M.Ma. carried out experiments and drafted the figures; L.G., D.L.T. and F.M.G. drafted and revised the tables; C.L., M.Me., M.G., G.G., I.N., C.M. and F.F. followed patients; L.C., M.S. and T.T. contributed to experimental procedures; S.D.B., L.G., D.L.T. and A.C. performed bioinformatic and statistical analyses; S.D.B., L.G., C.M., F.F. and A.C. conceived the study. All authors read and approved the paper.

## Competing interests
AQ, CP, GA, SD, DL, and JN are employers of Fluidigm Corporation; LC is the CEO at Labospace. All other authors declare no competing interest.
