## [Peer Review File · Nature Communications]

REVIEWER COMMENTS

Reviewer #1 (Remarks to the Author):

1) The authors state in the introduction that "Even though COVID-19 manifestation seems to be less severe in pregnant women than in elderly patients, it could be not completely absent, or silent." Do the authors have a specific reference in mind that substantiates this claim (which is a main rationale for the study)?

Of note, the references cited by the authors seem to indicate that pregnancy does not measurably influence COVID-19 symptom severity (such as ref 2, ref4). The authors state themselves that "In addition, clinical manifestations reported in pregnant women were mild and similar to those reported in non-pregnant women infected by SARS-CoV-2, with predominant features including fever, cough, dyspnea and lymphopenia (ref 6)". As such, what was the basis for the statement that COVID would be less severe in pregnancy?

2) Figure 1A and 1B are very dense and difficult to follow given the large number of separate panels. Readers would likely appreciate an additional heatmap and/or table in Figure 1 that summarizes (and groups) the cytokines by functional class, with additional columns specifying whether they are significantly altered in each of the pairwise comparisons.

3) What gestational ages are represented in this cohort? Do any of the measured parameters vary depending on gestational age?

4) Authors should incorporate some of the content in the Discussion into the Results section so that readers can place the data in context as they are reading it for the first time. For instance, the authors mention in the discussion that it is known that NK cells decrease in pregnancy, and that there is evidence that LDNs are induced in COVID-19. This is useful information to know while seeing the data. As a minor point, the (very brief) mention of the results shown in Figure 2C-D (NK cells and LDNs) should be moved up to the rest of the Results section where Figure 2 is introduced. Currently, the mention of NK cells and LDNs being different is placed after Figure 3, which makes for a confusing read. Also, it would be helpful to define what LDNs are on a cellular/functional level, since the markers that were used to define them are not particularly informative of their function.

5) The correlogram is a hard to read given the large number of tested parameters. I would suggest having additional panels that specifically show the relationship between the variables that are most correlated with LDN (seeing as that there are green boxes highlighting these). This could be simply shown with XY scatter plots for each variable (compared to LDN).

In addition, by comparing the top panel (pregnant negative) to the bottom panel (pregnancy positive), do the authors find differences in the directionality of correlation between LDN and these other variables? They mention that the positive correlations were "much more marked in SARS-CoV-2 infected pregnant women", but from eyeballing this figure, it appears that the directionality of the associations might even be inverted in uninfected pregnant women. In fact, in the discussion the authors write "In any case, the correlations between LDN and several soluble molecules, including inflammatory cytokines, that were present only in infected women deserve further attention." This should similarly be stated in the Results section, and there should be separate figure panels highlighting these differential correlations.

6) Given that the authors specifically measured these immune parameters in asymptomatic and paucisymptomatic patients, it is not particularly surprising that there were very few differences in

pregnant COVID+ vs pregnant COVID- patients. For instance, it is unlikely that most inflammatory cytokines would be measurably induced in asymptomatic COVID+ patients, regardless of pregnancy status. Nevertheless, I do think this study offers valuable insight and reassurance for pregnant patients who are asymptomatic but COVID+. However, if their goal was to argue that pregnancy leads to reduced inflammatory responses to COVID infection, they are missing an important 4th group of patients (nonpregnant, asymptomatic/paucisymptomatic COVID+ patients).

7) Overall comment -- the Results section currently reads like a bullet point list of data that were generated, with very limited interpretation or placing in context of the literature. This is likely a stylistic choice on the author's part, but as the manuscript currently is, the Results section is sparse while the Discussion section is extremely lengthy and long-winded. I would suggest the authors shuffle around some of the key concepts in the Discussion into the Results.

-- Ryan Chow and Sidi Chen

Reviewer #2 (Remarks to the Author):

OVERALL

The authors present a study profiling the immune system in SARS-CoV-2 and pregnancy. Though the cohort is small, which is acknowledged by the authors, they nonetheless provide a detailed and interesting insight into the immune dynamics in this unique context. In particular, the appearance of low density neutrophils in these asymptomatic or paucisymptomatic patients, despite relative stability of other immune subsets, is of particular interest. I enjoyed reading this study, and I commend the effort here to study a relatively under-investigated space!

There are a few areas where I believe some additional data or information would strengthen, or clarify, the arguments made.

MAJOR

A number of the following point might be self evident from an examination of the data. Unfortunately I was not able to access the files on Flow Repository using the number provided (FR-FCM-Z3GH). I would suggest that the authors make the files available (even if through another means, such as Dropbox, for review). In the meantime, I have listed my points here:

1. Where the mass cytometry samples stained and run in multiple batches? If so, it is possible that there could be shifts between samples that are due to technical, rather than biological variance. In some analysis this can lead to population shifts between clusters. Some addition plots in the supplementary data coloured by batch (and/or patient) would be indicative, and would greatly clarify the population positions here.

2. Additionally, some supplementary UMAP plots coloured by the expression of each marker would be critical in assessing the population labelling performed via annotated FlowSOM clusters.

3. Line 142: "Two clusters, expressing or not CD11c, represented DC, identified by the expression of CD123". I think the CD11c+CD123+ cells are likely to be plasmacytoid DCs, and the CD11c-CD123+ could possibly be basophils. Though I note that these are PBMCs, and that these have had clean-up gating, it is still possible basophils have persisted. Notably, the CD11c-CD123+ cluster (pale brown) is very distinct from the CD11c+CD123+ population (purple), which appears to closely cluster with the

monocytes (teal). Interestingly, both of the nominal DC populations express CD4 (but not CD3), which does suggest a plasmacytoid DC phenotype, possibly instead of one being basophils.

4. Line 143: "Mucosal associated invariant T cells (MAIT) were identified by the expression of CD161". I believe this should also be expressed on NK cells, though the 'MAIT' population in Figure 2B appears to express CD3 and not CD56, so perhaps the text could be altered to reflect this.

5. Line 158: "The percentage of LDN was higher in pregnant women with COVID-19 compared to pregnant women without infection (Figure 2C, green arrow). ". It seems that in Figure 2C, for the LDN, these are significantly different in HD vs PRE-, HD vs PRE+, PRE-vs PRE+, though only one star for the LDN plot is seen in Figure 2D (HD vs PRE-). Were different statistics used? Elevated LDN in PRE+ cells in PBMCs is consistent with responses to COVID-19, but I am interested in the apparent elevation of these cells in PRE- vs HD, suggesting some form of inflammatory pull on the bone marrow to release immature neutrophils (unless the low density neutrophil phenomena is due to some other cause in pregnancy that is unfamiliar to me).

6. Line 161 onwards: some of the chemokine gates are a little tricky to interpret, as very few positive cells are apparent (e.g. Supp Figure 1 -- CCR4 vs GATA3). Is it possible these are these same because there aren't any cells expressing these markers?

MINOR

1. Figure 2: the methods text indicates that the numerical data in Figure 2C and Figure 2D are percentage of live cells, but adding this to the legend (or the figure itself) would be helpful

2. Figure 3 labels for the plots would be helpful

3. Line 235: the direction of change for the intermediate monocytes should be stated

Thomas Ashhurst

Reviewer #3 (Remarks to the Author):

In this study by Biasi et al. , the authors identify changes in immunological markers, cytokines/chemokines, between nonpregnant/SARS-CoV-2 uninfected pregnant women/ SARS-CoV-2 infected pregnant women. The authors explored the plasma levels of cytokines and chemokines and related them to the innate and adaptive immunity cells within PBMC. Interestingly, the authors found increased expression of inflammatory cytokines, specifically IL-1RA, IL-10, IL-19, and low IL-17, PDL-1, and D-dimer levels. The changes in the profile of T and B cells did not change after infection that suggests that SARS-CoV-2 infection cannot alter the delicate equilibrium that regulates the immunological tolerance. Overall the authors have provided detailed information and immunological features in pregnant women uninfected and infected with SARS-CoV-2.

Main concerns-

1. In the introduction, the authors only include studies that do not show any severe outcome after SARS-CoV-2 infection in pregnant women. However, multiple reports now indicate the possibility of vertical transmission, fetal demise , pregnancy complications, and maternal death due to COVID-19 infection. The authors should provide a broad overall picture of the disease then narrow down immunological features after infection during pregnancy.

2. In Fig 1A, the expression of PDGF-AA does not appear to be increasing after infection compared with uninfected pregnant women.
3. some of the cytokines mentioned in Fig1A and 1B show opposite patterns like IL-1RA, IL-27, CCL4, APRIL, BAFF.
4. In Fig 1B, the chemokine CXCL12 is missing.
5. The study suggests that pregnant women can dampen inflammation because of the immunosuppressive environment of pregnancy. It would be nice to compare immune responses between pregnant COVID patients and nonpregnant COVID patients.
6. What is the significance of EGF & PDGF-AA; It would be nice to include descriptions explaining what these molecules do and explain why they might be elevated in COVID patients.
7. Lines 275-281, what could be the reason for PDL-1 upregulation and downregulation in COVID-19 negative and positive women, respectively.

Minor comments

1. Line 80-81, reference is missing.
2. Line 174-175, the sentence is not clearly written.

POINT TO POINT RESPONSE TO THE COMMENTS OF THE REVIEWERS

Reviewer#1(Remarks to the Author):

1) The authors state in the introduction that "Even though COVID-19 manifestation seems to be less severe in pregnant women than in elderly patients, it could be not completely absent, or silent." Do the authors have a specific reference in mind that substantiates this claim (which is a main rationale for the study)?

Of note, the references cited by the authors seem to indicate that pregnancy does not measurably influence COVID-19 symptom severity (such as ref 2, ref 4). The authors state themselves that "In addition, clinical manifestations reported in pregnant women were mild and similar to those reported in non-pregnant women infected by SARS-CoV-2, with predominant features including fever, cough, dyspnea and lymphopenia (ref 6)". As such, what was the basis for the statement that COVID would be less severe in pregnancy?

We agree and thank the reviewer for raising these questions, that allowed us to revise the entire introduction citing new references, related to papers that report percentages of infection and symptoms in pregnant women with SARS.-CoV-2 infection.

2) Figure 1A and 1B are very dense and difficult to follow given the large number of separate panels. Readers would likely appreciate an additional heatmap and/or table in Figure 1 that summarizes (and groups) the cytokines by functional class, with additional columns specifying whether they are significantly altered in each of the pairwise comparisons.

According to this comment, Figure 1A and 1B have been replaced by Table 1 that reports the mean and standard deviation for each soluble molecule. Molecules are also organized by their functional class and p values are reported.

3) What gestational ages are represented in this cohort? Do any of the measured parameters vary depending on gestational age?

As reported in supplementary table 1, the gestational ages are the following:

- Pregnant positive: median 30; mean 24.74; SD of the mean 14.43.
- Pregnant negative: median 39; mean 33; SD of the mean 14.69.

The cohort included and analyzed has been matched for age and gestational ages and none of the differences shown in the measured parameters vary depending on gestational ages.

4) Authors should incorporate some of the content in the Discussion into the Results section so that readers can place the data in context as they are reading it for the first time. For instance, the authors mention in the discussion that it is known that NK cells decrease in pregnancy, and that there is evidence that LDNs are induced in COVID-19. This is useful information to know while seeing the data. As a minor point, the (very brief) mention of the results shown in Figure 2C-D (NK cells and LDNs) should be moved up to the rest of the Results section where Figure

2 is introduced. Currently, the mention of NK cells and LDNs being different is placed after Figure 3, which makes for a confusing read. Also, it would be helpful to define what LDNs are on a cellular/functional level, since the markers that were used to define them are not particularly informative of their function.

We thank the referee for this suggestion. This section was amended as indicated, and most paragraphs have been entirely revised.

5) The correlogram is a hard to read given the large number of tested parameters. I would suggest having additional panels that specifically show the relationship between the variables that are most correlated with LDN (seeing as that there are green boxes highlighting these). This could be simply shown with XY scatter plots for each variable (compared to LDN).

In addition, by comparing the top panel (pregnant negative) to the bottom panel (pregnancy positive), do the authors find differences in the directionality of correlation between LDN and these other variables? They mention that the positive correlations were "much more marked in SARS-CoV-2 infected pregnant women", but from eyeballing this figure, it appears that the directionality of the associations might even be inverted in uninfected pregnant women. In fact, in the discussion the authors write "In any case, the correlations between LDN and several soluble molecules, including inflammatory cytokines, that were present only in infected women deserve further attention." This should similarly be stated in the Results section, and there should be separate figure panels highlighting these differential correlations.

We considered this useful suggestion, and we amended the figure by adding a panel on the left of the pregnant positive correlogram, showing XY scatter plots of the most relevant variables correlated with LDN. We also added the following paragraph in the discussion: "The correlations here described between different cytokines or between a given cytokine and hematochemical and immunological parameters are quite complex. It is interesting to note that in infected women, but not in pregnant negative women, LDN levels show significant correlations with a number of cytokines and soluble molecules that have different functions and exert different effects on these cells. For example, several functional activities of polymorphonuclear neutrophils are subject to regulation by immunosuppressive molecules like IL-4, IL10 and IL-13, but also to the modulation by molecules favoring their proinflammatory capacity such as IL-1, IL-17, IL-18 and different interferons. It has been shown two major populations of LDN exist, such as immunosuppressive LDN, also known as polymorphonuclear myeloid-derived suppressor cells, that are typically found in cancer, pregnancy, infections, and systemic inflammation 42-44 and proinflammatory LDN, or low-density granulocytes, that are typically found in autoimmune diseases. At present, standardized markers are not yet available to define such populations, and their pro- or anti-inflammatory function can be only verified by testing in vitro their activity 45. On the one side, considering that several cytokines quoted here can also be produced by cells of the granulocytic family, the correlations of LDN with cytokines with different activity could be due to the existence of the two cell populations described above. On the other, even if we could not measure any functional parameter, it can be hypothesized that, in our cohort of pregnant women infected by SARS-CoV-2, LDN

with immunosuppressive capabilities were present, and were able to control inflammation.”

6) Given that the authors specifically measured these immune parameters in asymptomatic and paucisymptomatic patients, it is not particularly surprising that there were very few differences in pregnant COVID+ vs pregnant COVID- patients. For instance, it is unlikely that most inflammatory cytokines would be measurably induced in asymptomatic COVID+ patients, regardless of pregnancy status. Nevertheless, I do think this study offers valuable insight and reassurance for pregnant patients who are asymptomatic but COVID+. However, if their goal was to argue that pregnancy leads to reduced inflammatory responses to COVID infection, they are missing an important 4th group of patients (nonpregnant, asymptomatic/paucisymptomatic COVID+ patients).

We acknowledge the reviewer for this comment. We totally agree that the inclusion of a 4th group of patients such as nonpregnant, asymptomatic/paucisymptomatic COVID+ patients would have improved and highlighted the importance of the response to COVID infection by pregnant women.

However, unfortunately it is quite complex to recruit this group of persons. Indeed, asymptomatic/paucisymptomatic COVID+ patients typically arrive to the hospital for a test after a suspicious contact, receive the nasal swab, and are immediately sent home, where they receive the response after a few hours. Then, according to Italian law, if positive and asymptomatic or paucisymptomatic they must stay in quarantine, and cannot be visited unless strictly necessary. No one can have contact with them, and after the indicated quarantine period they must go to specific drive-in points to perform the nasopharyngeal swab. For these reasons, it is almost impossible for us to collect blood of these patients. Nonetheless, cytokines in asymptomatic individuals have been in deep investigated here (<https://doi.org/10.1038/s41591-020-0965-6>). Serum cytokines and chemokines levels were compared between the asymptomatic and symptomatic groups. Elevated concentrations of 18 pro- and anti-inflammatory cytokines were observed in the symptomatic group as compared to the asymptomatic group. Of these, tumor necrosis TRAIL, M-CSF, GRO- α , G-CSF and IL-6 showed the most significant changes. Moreover, the cytokines were further analyzed in the asymptomatic group and the 37 healthy controls. The plasma levels of 32 cytokines were similar between the healthy controls and the asymptomatic individuals. Significantly higher levels of SCF, IL-13, IL-12 p40 and LIF were found in the asymptomatic group. Collectively, these data show that the asymptomatic individuals had a reduced inflammatory response characterized by low circulating concentrations of cytokines and chemokines.

Moreover, high frequency of NK cells and early and transient increase of specific IgA, IgM and, to a lower extent, IgG are associated with asymptomatic SARS-CoV-2 (doi: 10.3389/fimmu.2020.610300). No data regarding T and B cell immunophenotype of asymptomatic infected individuals have been published by now. In fact, T cell immunophenotype and response have been evaluated in convalescent individuals with asymptomatic or mild COVID-19 (<https://doi.org/10.1016/j.cell.2020.08.017>) and the immune profiling of COVID-19 hospitalized patients have been investigated (<https://doi.org/10.1038/s41421-021-00250-9>) (<https://doi.org/10.1126/science.abc851>).

Unpublished data by Antonio Bertoletti lab revealed that comparing a cohort of asymptomatic individuals (n=85) with that of symptomatic COVID-19 patients (n=76), the asymptomatic SARS-CoV-2 infected individuals (n=85) mount a robust and highly functional virus-specific cellular immune response (<https://doi.org/10.1101/2020.11.25.399139>). Overall, these data suggest that asymptomatic/ paucisymptomatic patients mount an immune response after COVID-19 infection.

7) Overall comment -- the Results section currently reads like a bullet point list of data that were generated, with very limited interpretation or placing in context of the literature. This is likely a stylistic choice on the author's part, but as the manuscript currently is, the Results section is sparse while the Discussion section is extremely lengthy and long-winded. I would suggest the authors shuffle around some of the key concepts in the Discussion into the Results.

Thank you for this final comment. The result section and the discussion have been deeply revised, as suggested.

-- Ryan Chow and Sidi Chen

Reviewer #2 (Remarks to the Author):

OVERALL

The authors present a study profiling the immune system in SARS-CoV-2 and pregnancy. Though the cohort is small, which is acknowledged by the authors, they nonetheless provide a detailed and interesting insight into the immune dynamics in this unique context. In particular, the appearance of low-density neutrophils in these asymptomatic or paucisymptomatic patients, despite relative stability of other immune subsets, is of particular interest. I enjoyed reading this study, and I commend the effort here to study a relatively under-investigated space!

We thank the reviewer for the appreciation.

There are a few areas where I believe some additional data or information would strengthen, or clarify, the arguments made.

We amended all the points as requested.

MAJOR

A number of the following point might be self evident from an examination of the data. Unfortunately I was not able to access the files on Flow Repository using the number provided (FR-FCM-Z3GH). I would suggest that the authors make the files available (even if through another means, such as Dropbox, for review). In the meantime, I have listed my points here:

We apologize for the inconvenience, but something went wrong in making public the experiments and the folder containing FCS files. As soon as we received the letter from

the Editor with the comments, we fixed the problem and indeed the day after we wrote to the Editor with the request to pass the fixed and working link to all the reviewers.

1. Where the mass cytometry samples stained and run in multiple batches? If so, it is possible that there could be shifts between samples that are due to technical, rather than biological variance. In some analysis this can lead to population shifts between clusters. Some additional plots in the supplementary data coloured by batch (and/or patient) would be indicative, and would greatly clarify the population positions here.

We thank the reviewer for this comment that allow us to clarify the issue of data reproducibility and the importance of batch effect in this kind of unsupervised analyses. Samples were run in five different days. Batch effect was checked sample by sample, day by day and we excluded that technical variance is responsible of shift between clusters. As suggested, to clarify this point, we added two supplementary figures. UMAP plots colored by day of acquisition and showing the cell distribution are shown in supplementary figure 4. UMAP plots showing cell and cluster distribution of single patients are shown in supplementary figure 5.

2. Additionally, some supplementary UMAP plots coloured by the expression of each marker would be critical in assessing the population labelling performed via annotated FlowSOM clusters.

We amended this request and supplementary figure 6, showing maker distribution across UMAP projection, has been added.

3. Line 142: "Two clusters, expressing or not CD11c, represented DC, identified by the expression of CD123". I think the CD11c+CD123+ cells are likely to be plasmacytoid DCs, and the CD11c-CD123+ could possibly be basophils. Though I note that these are PBMCs, and that these have had clean-up gating, it is still possible basophils have persisted. Notably, the CD11c-CD123+ cluster (pale brown) is very distinct from the CD11c+CD123+ population (purple), which appears to closely cluster with the monocytes (teal). Interestingly, both of the nominal DC populations express CD4 (but not CD3), which does suggest a plasmacytoid DC phenotype, possibly instead of one being basophils.

We thank the reviewer for raising this thought that allows as to investigate more in depth the classification we made on the basis of markers expression. Plasmacytoid DCs are classified as CD11c-C123+, while the CD11c+C123+ DCs represent DCs progenitors (*Alcántara-Hernández, Marcela, et al. "High-dimensional phenotypic mapping of human dendritic cells reveals interindividual variation and tissue specialization." Immunity 47.6 (2017): 1037-1050.*) CD11c+C123+ DCs could not be basophils as they do not express CD11b (*Chevrier, Stéphane, et al. "A distinct innate immune signature marks progression from mild to severe COVID-19." Cell Reports Medicine 2.1 (2021): 100166.*)

Accordingly, we amended the text and also the figure legend of revised figure 1: DCs CD11b- has been replaced by pDC while DCs CD11c+ has been replaced by "DC progenitors".

4. Line 143: "Mucosal associated invariant T cells (MAIT) were identified by the expression of CD161". I believe this should also be expressed on NK cells, though the 'MAIT' population in Figure 2B appears to express CD3 and not CD56, so perhaps the text could be altered to reflect this.

The reviewer is correct. Thus, we modify the sentence:

"MAIT were defined by the expression of CD3, CD161 and the lack of expression of CD56, and similar percentages were present in the different groups analyzed."

5. Line 158: "The percentage of LDN was higher in pregnant women with COVID-19 compared to pregnant women without infection (Figure 2C, green arrow). ". It seems that in Figure 2C, for the LDN, these are significantly different in HD vs PRE-, HD vs PRE+, PRE-vs PRE+, though only one star for the LDN plot is seen in Figure 2D (HD vs PRE-). Were different statistics used? Elevated LDN in PRE+ cells in PBMCs is consistent with responses to COVID-19, but I am interested in the apparent elevation of these cells in PRE- vs HD, suggesting some form of inflammatory pull on the bone marrow to release immature neutrophils (unless the low density neutrophil phenomena is due to some other cause in pregnancy that is unfamiliar to me).

The asterisk in figure 2D did not express the p-value, but was used to underline the main changes reported in the figure 2C. However, to avoid any miss-interpretations, we have changed the type of graphs of figure 2D. In the actual figure, the median and interquartile range are plotted together with the p-value of statistical relevant differences observed. The presence of LDN within healthy pregnant woman has already been reported in literature (*Kropf, P. et al. "Arginase activity mediates reversible T cell hyporesponsiveness in human pregnancy." European journal of immunology 37.4 (2007): 935-945. and Ssemaganda, Aloysius, et al. "Characterization of neutrophil subsets in healthy human pregnancies." PloS one 9.2 (2014): e85696.*). In normal pregnancy, there is an increased systemic inflammation, enhanced number of polymorphonuclear cells, a low Th1/Th2 balance, a decrease in peripheral NK cells, and an increased number of regulatory T cells. As suggested by the reviewer, it is possible that, due to pregnancy and increased number of neutrophils, immature form of neutrophils could be detected in the circulation, such as LDN. Those cells display an anti-inflammatory capacity mediated by the expression of arginase that cause a depletion of L-arginine and consequently the downregulation of lymphocyte responses. This could be linked to the immune suppressive environment observed within pregnant woman.

6. Line 161 onwards: some of the chemokine gates are a little tricky to interpret, as very few positive cells are apparent (e.g. Supp Figure 1 -- CCR4 vs GATA3). Is it possible these are these same because there aren't any cells expressing these markers?

Thanks for this comment. In order to investigate chemokine expression within CD4 and CD8 T cell, we applied Poisson statistic to calculate the number of events to acquire in order to detect a population represented about 2% in healthy donors (about 200,000 viable

cells). Where the number of events were not enough, we acquired all the events possible. In addition, FMO were used to check gates as we admit that some gates were difficult to set. In conclusion, we can assert that we did not find any differences in the marker expression.

MINOR

1. Figure 2: the methods text indicates that the numerical data in Figure 2C and Figure 2D are percentage of live cells, but adding this to the legend (or the figure itself) would be helpful

We amended as requested and we add to the figure legend the following sentence: "The dot plots show the relative abundancies of 27 population found within pre-cleaned CD45+ live cells. Values (dots) for the three conditions are matched with the color used in C. Data represent individual percentage values (dots), median (center bar) and interquartile range (whiskers)."

2. Figure 3 labels for the plots would be helpful

We amended as suggested and labels were added to figure 3, now revised figure 2.

3. Line 235: the direction of change for the intermediate monocytes should be stated

We amended as requested and we modified as follow:

"The expression of CD16 and CD14 allows the identification of three types of monocytes, i.e. classical, intermediate and non-classical. These cells are modified during pregnancy: the sub-population of intermediate monocytes increases, while classical monocytes decrease and there are no changes in the non-classical subpopulation²³. Here, we show that all monocyte populations were not phenotypically different between infected or non-infected women, indicating that likely these cells do not participate to the creation of an inflammatory milieu"

Thomas Ashhurst

Reviewer #3 (Remarks to the Author):

In this study by Biasi et al., the authors identify changes in immunological markers, cytokines/chemokines, between nonpregnant/SARS-CoV-2 uninfected pregnant women/ SARS-CoV-2 infected pregnant women. The authors explored the plasma levels of cytokines and chemokines and related them to the innate and adaptive immunity cells within PBMC. Interestingly, the authors found increased expression of inflammatory cytokines, specifically IL-1RA, IL-10, IL-19, and low IL-17, PDL-1, and D-dimer levels. The changes in the profile of T and B cells did not change after infection that suggests that SARS-CoV-2 infection cannot alter the delicate equilibrium that regulates the immunological tolerance. Overall, the authors have provided detailed information and

immunological features in pregnant women uninfected and infected with SARS-CoV-2.

Main concerns

1. In the introduction, the authors only include studies that do not show any severe outcome after SARS-CoV-2 infection in pregnant women. However, multiple reports now indicate the possibility of vertical transmission, fetal demise, pregnancy complications, and maternal death due to COVID-19 infection. The authors should provide a broad overall picture of the disease then narrow down immunological features after infection during pregnancy.

We acknowledge the referee for the comment. The introduction has been completely reworked providing a broad overall picture of COVID-19 infection during pregnancy, including the mention of severe cases among pregnant women.

2. In Fig 1A, the expression of PDGF-AA does not appear to be increasing after infection compared with uninfected pregnant women.

Figure 1A and B have been removed and it has been replaced with a more readable table, now Table 1. Mean and standard deviation, together with p value are now reported.

3. some of the cytokines mentioned in Fig1A and 1B show opposite patterns like IL-1RA, IL-27, CCL4, APRIL, BAFF.

The paragraph has been revised and corrected, accordingly.

4. In Fig 1B, the chemokine CXCL12 is missing.

Thank you for pointing out this mistake, that has been corrected as suggested.

5. The study suggests that pregnant women can dampen inflammation because of the immunosuppressive environment of pregnancy. It would be nice to compare immune responses between pregnant COVID patients and nonpregnant COVID patients.

We thank the reviewer for this comment that allow us to explain why a fourth group of patients is missing. As reported in the reply to the comment of Review #1, asymptomatic or paucisymptomatic patients do not have access to the hospital, and they are forced to go to home quarantine. According to Italian law, no one can have contact with them, and they must go to specifically set up drive-in points to perform the nasopharyngeal swab. For these reasons, it is almost impossible to collect blood from these patients.

6. What is the significance of EGF & PDGF-AA; It would be nice to include descriptions explaining what these molecules do and explain why they might be elevated in COVID patients.

We thank the reviewer for this point that allowed us to better describe the role of different molecules. The paragraph has been revised and a description of EGF and PDGF-AA together with their role have been included and reported below:

“PDGF-AA is a plasma factor responsible of vascular remodeling, and recently published data report that COVID-19 infected patients have increased levels of molecules involved in this phenomenon (CD40L, PDGF-AA, PDGF-AB/BB) that correlate with high level of the

Th2 cytokine IL-4¹⁵. EGF is involved in cellular proliferation, differentiation, survival, and can modulate the wound healing response to SARS-CoV¹⁶. Moreover, COVID-19 infection itself leads to the activation of growth factor receptor signaling, maybe sustaining a more marked EGF production¹⁷. Besides, the levels of PDGF-AA and EGF correlate with the severity of the disease, and high levels of angiogenesis factors were elevated in hospitalized patients with non-critical COVID-19 infection¹⁸.”

7. Lines 275-281, what could be the reason for PDL-1 upregulation and downregulation in COVID-19 negative and positive women, respectively.

This is an important question indeed. Levels of soluble forms of PD-L1 (sPD-L1) were reported to be higher in the serum of pregnant women than in non-pregnant women. sPD-L1 is able to suppress maternal immunity, inhibiting monocyte and T cells function (Okuyama M, Mezawa H, Kawai T and Urashima M (2019) Elevated Soluble PD-L1 in Pregnant Women's Serum Suppresses the Immune Reaction. *Front. Immunol.* 10:86. doi: 10.3389/fimmu.2019.00086). Low level of sPD-L1 has been found in asymptomatic COVID-19 patients (Kong, Y., Wang, Y., Wu, X. et al. Storm of soluble immune checkpoints associated with disease severity of COVID-19. *Sig Transduct Target Ther* 5, 192 (2020). <https://doi.org/10.1038/s41392-020-00308-2>).

Minor comments

1. Line 80-81, reference is missing.

The introduction has been completely revised and the sentence has been removed.

2. Line 174-175, the sentence is not clearly written.

The sentence has been removed because the indicated difference (percentage of CD161+,CCR6+ T cells) was not significant.

REVIEWERS' COMMENTS

Reviewer #1 (Remarks to the Author):

revision addressed prior comments.

Ryan Chow and Sidi Chen

Reviewer #2 (Remarks to the Author):

The edits made by the authors have satisfied the items raised in my initial review. I have no further concerns and consider this suitable for publication.

I would like to acknowledge the authors on a very interesting study!

- Thomas AShurst